# Learning Policy Committees for Effective Personalization in MDPs with Diverse Tasks

## Abstract

Many dynamic decision problems, such as robotic control, involve a series of tasks, many of which are unknown at training time. Typical approaches for these problems, such as multi-task and meta reinforcement learning, do not generalize well when the tasks are diverse. We propose a general framework to address this issue. In our framework, the goal is to learn a set of policies—a *policy committee*—such that at least one is near-optimal for most tasks that may be encountered at execution time. While we show that even a special case of this problem is inapproximable, we present two effective algorithmic approaches for it. The first of these yields provably approximation guarantees, albeit in low-dimensional settings (the best we can do due to inapproximability), whereas the second is a general and practical gradient-based approach. In addition, we provide provable sample complexity bounds for few-shot learning settings. Our experiments in personalized and multi-task RL settings on MuJoCo and Meta-World show that the proposed approach outperforms state-of-the-art multi-task, meta-, and personalized RL baselines on training and test tasks, as well as in few-shot learning, often by a large margin.

## 1 Introduction

Reinforcement learning (RL) has achieved remarkable success in a variety of domains, from robotic control (Lillicrap, 2015) to game playing (Xu et al., 2018). However, many real-world applications involve highly diverse sets of tasks, making it impractical to rely on a single, fixed policy. In these settings, both the reward structures and the transition dynamics can vary significantly across tasks. Existing approaches to this challenge—such as multi-task RL (MTRL) and meta-reinforcement learning (meta-RL)—struggle to generalize effectively when tasks are both diverse and previously unseen.

Multi-task RL methods typically train a single policy or a shared representation across tasks (Vithayathil Varghese & Mahmoud, 2020). However, they often face negative transfer, where optimizing for one task degrades performance on others (Zhang et al., 2022). This limitation becomes more pronounced when tasks require drastically different strategies, as the policy is forced to handle conflicting objectives. On the other hand, meta-RL approaches, such as Model-Agnostic Meta-Learning (MAML) (Finn et al., 2017b) and PEARL (Rakelly et al., 2019), aim to enable fast adaptation to new tasks but rely heavily on fine-tuning at test time, which can be computationally expensive and ineffective in environments with high variability in both rewards and transitions, like Meta-World benchmark tasks (Yu et al., 2020c). Furthermore, these methods typically underperform on the training tasks compared to MTRL due to the generalization trade-off inherent in meta-learning. Another promising direction is personalized RL (Ivanov & Ben-Porat, 2024), where multiple policies are trained to cater to diverse reward structures, but this line of work focuses predominantly on rewards while assuming shared transition dynamics.

According to *script theory*, human cognition enables effective generalization in an open world by learning a collection of *scripts*, or behavioral patterns (Abelson, 1981; Schank, 1983), one of which can then be selected and adapted as needed to particular predicaments. Inspired by this concept, we propose a novel **policy committee framework** designed to efficiently handle environments with **diverse task distributions**, where both reward functions and transition dynamics can vary significantly across tasks. Instead of learning a single policy or relying on complex fine-tuning, our

framework learns a set of policies—a committee—where each policy is specialized to handle a specific subset of tasks. This allows for task-specific expertise while maintaining generalization across a wide range of task variations. We refer to our approach as PACMAN.

To summarize, our key contributions are as follows:

- **Simple and Effective Policy Committees**: Our policy committee approach scales efficiently with task diversity. By learning a small set of policies that cover a broad range of tasks, we reduce the computational complexity compared to methods that require a separate policy for each task or complex adaptation procedures. Additionally, our approach leverages LLM-based task embeddings for non-parametric tasks, which provides a more general and scalable solution to environments where tasks cannot be easily parameterized.
- **Theoretical Analysis:** We first provide a general computational impossibility result, showing that even the problem of identifying the optimal sets of tasks for policy committee training is inapproximable. However, we also present an efficient algorithmic approach with worst-case approximation guarantees in the special case when task embedding dimension is constant, and a general gradient-based approach, albeit with weaker guarantees. Finally, we theoretically demonstrate few-shot efficacy of our approach by showing that it has sample complexity that is linear in the size of the committee and *independent of the size of the state and action space*.
- **Empirical Validation:** We demonstrate the efficacy of the proposed PACMAN approach through extensive experiments on challenging multi-task benchmarks, including MuJoCo and Meta-World. Our policy committee framework consistently outperforms state-of-the-art multi-task RL and meta-RL baselines in both zero-shot and few-shot settings, achieving better generalization and faster adaptation across diverse tasks.

**Related Work:** Our work is closely related to three key areas within the broader reinforcement learning literature: multi-task RL, personalized RL, and meta-RL.

*Multi-Task RL (MTRL):* A major advantage of MTRL over single-task learning is the ability to share knowledge across tasks, a concept extensively explored in various studies proposing different methods to utilize task relationships (Yang et al., 2020b; Sodhani et al., 2021; Sun et al., 2022). However, naive knowledge-sharing across tasks can lead to negative transfer, as not all tasks benefit from shared knowledge. Consequently, learning a task-specific skill may distract from the learning of other tasks. A notable area of research examines task interference in MTRL through the lens of gradient alignment. Yu et al. (2020a) tackles it by projecting the gradient of a task to the orthogonal direction of all the other tasks, while Hessel et al. (2019) addresses it via synchronizing the gradient magnitude across tasks. Numerous methods in the literature aim to address task interference issues from a representation learning perspective. Sodhani et al. (2021) learn a mixture of state encoders shared across tasks, that helps generate diverse representations through an attention mechanism. Lan et al. (2024) introduce the Contrastive Modules with Temporal Attention (CMTA) framework, which leverages contrastive learning to ensure the modules are distinct from one another and integrates shared modules at a finer granularity than the task level using temporal attention. Recently, Hendawy et al. (2023) proposed an approach called Mixture of Orthogonal Experts (MOORE) that captures common structures among tasks by employing orthogonal representations to enhance diversity. MOORE utilizes a Gram-Schmidt process to create a shared subspace of representations derived from a mixture of experts. While all these previous MTRL approaches focus on learning a policy to efficiently address a predefined set of tasks, our focus is to learn a set of policies such that at least one policy in the set is near-optimal for most *previously unseen tasks*.

*Personalized-RL:* Recently, Ivanov & Ben-Porat (2024) introduced personalized RL, to accommodate a diverse user population, each with distinct preferences, through interaction with a small set of representative policies. Although the personalized RL framework has some similarities to our approach, we adopt a broader setup allowing for variations both in rewards and transition dynamics, since many real-world scenarios warrant mastering a diverse set of tasks that comprise different dynamics. Moreover, we empirically show that PACMAN significantly outperforms the state-of-the-art personalized RL method across diverse evaluation settings even when only rewards vary.

*Meta-RL:* Meta-RL methods can be categorized broadly into two categories, (i) context-based; and (ii) gradient-based. Context-based methods primarily rely on learning a context (Bing et al., 2023; Gupta et al., 2018; Duan et al., 2017; Lee et al., 2020a;b; 2023; Rakelly et al., 2019) by employing RNN or LSTM-based neural networks to encode collected experiences into a latent context embedding, and then act by conditioning the policy on the learned context. However, they are susceptible

to distribution shifts at inference time, as the encoded context and the policy derived from that context often struggle to generalize to out-of-distribution tasks. Additionally, the parameters of the latent context encoder are trained to predict reward and/or transition dynamics based on the context, typically involving the minimization of a KL divergence-based loss. Consequently, the learned context tends to exhibit mode-seeking behavior, which poses a significant limitation in situations that require capturing diverse, multi-modal context (such as in Meta-World). Recently, Bing et al. (2023) attempt to address this issue in non-parametric tasks by using task-specific detailed natural language instructions. Several gradient-based methods (Finn et al., 2017b; Stadie et al., 2018; Mendonca et al., 2019; Zintgraf et al., 2019) have been developed to address the few-shot adaptation challenge. These approaches focus on learning a shared initialization of a model across tasks, allowing the agent to achieve strong performance on unseen target tasks with only a few gradient updates. These approaches are not well-suited for zero-shot generalization problems, as they typically require numerous gradient steps through the policy to learn an effective policy for a given task. Finally, since meta-RL methods prioritize rapid adaptation, they often fall short of state-of-the-art MTRL performance on in-sample (training) tasks. In this work, we aim to close this gap by developing a framework that excels in both in-sample and out-of-sample tasks.

## 2 MODEL

We consider the following general model of *multi-task MDPs (MT-MDP)*. Suppose we have a *dynamic environment* $\mathcal{E} = (\mathcal{S}, \mathcal{A}, h, \gamma, \rho)$ where $\mathcal{S}$ is a state space, $\mathcal{A}$ an action space, $h$ the decision horizon, $\gamma$ the discount factor, and $\rho$ the initial state distribution. Let a *task* $\tau = (\mathcal{T}, r)$ in which $\mathcal{T}$ is the transition model where $\mathcal{T}(s, a)$ is a probability distribution over next state $s'$ as a function of current state-action pair $(s, a)$ and $r(s, a)$ the reward function. A Markov decision process (MDP) is thus a composition of the dynamic environment and task, $(\mathcal{E}, \tau)$.

Let $\Gamma$ be a distribution over tasks $\tau$. We define a *MT-MDP* $\mathcal{M}$ as the tuple $(\mathcal{E}, \Gamma)$, as in typical meta-RL models (Beck et al., 2023; Wang et al., 2024). Additionally, we define a *finite-sample* variant of MT-MDP, *FS-MT-MDP*, as $\mathcal{M}_n = (\mathcal{E}, \tau_1, \ldots, \tau_n)$, where $\tau_i \sim \Gamma$. An FS-MT-MDP thus corresponds to multi-task RL (Zhang & Yang, 2021).

At the high level, our goal is to learn a *committee of policies* $\Pi = \{\pi_1, \ldots, \pi_K\}$ such that for most tasks, there exists at least one policy $\pi \in \Pi$ that is effective. Next, we formalize this problem. Let $V_\tau^\pi$ be the value of a policy $\pi$ for a given task $\tau$, i.e.,

$$V_\tau^\pi = \mathbb{E}\left[\sum_{t=0}^{h} \gamma^t r_\tau(s_t, a_t) | a_t = \pi(s_t)\right],$$

where the expectation is with respect to $\mathcal{T}_\tau$ and $\rho$. Let $V_\tau^*$ denote an optimal policy for a task $\tau$. Define $V_\tau^\Pi = \max_{\pi \in \Pi} V_\tau^\pi$, that is, we let the value of a committee $\Pi$ to a task $\tau$ be the value of the *best* policy in the committee for this task. There are a number of reasons why this evaluation of a committee is reasonable. As an example, if a policy implements responses to prompts for conversational agents and $\Pi$ is small, we can present multiple responses if there is significant semantic disagreement among them, and let the user choose the most appropriate. In control settings, we can rely on domain experts who can use additional semantic information associated with each $\pi \in \Pi$ and the tasks, such as the descriptions of tasks $\pi$ was effective for at training time, and similar descriptions to test-time tasks, to choose a policy. Moreover, as we show in Section 4, this framework leads naturally to effective few-shot adaptation, which requires neither user nor expert input to determine the best policy.

One way to define the value of a policy committee $\Pi$ with respect to a given MT-MDP and FS-MT-MDP is, respectively, as $V_\mathcal{M}^\Pi = \mathbb{E}_{\tau \sim \Gamma}\left[V_\tau^\Pi\right]$ and $V_{\mathcal{M}_n}^\Pi = \frac{1}{n}\sum_{i=1}^{n} V_{\tau_i}^\Pi$. The key problem with these learning goals is that when the set of tasks is highly diverse, different tasks can confound learning efficacy for one another. For example, if we have several groups of tasks such that within-group tasks are quite similar to one another, but with tasks differing significantly (e.g., requiring fundamentally different skills) across groups, learning a single policy that is effective for all tasks will be extremely challenging, with tasks from different groups sending conflicting reward signals.

We address this limitation by defining the goal of policy committee learning differently. First, we formalize what it means for a committee $\Pi$ to have a *good* policy for *most* of the tasks.

**Definition 1.** *A policy committee $\Pi$ is an $(\epsilon, 1 - \delta)$-cover for a task distribution $\Gamma$ if $V_\tau^\Pi \geq V_\tau^* - \epsilon$ with probability at least $1 - \delta$ with respect to $\Gamma$. $\Pi$ is an $(\epsilon, 1 - \delta)$-cover for a set of tasks $\{\tau_1, \ldots, \tau_n\}$ if $V_\tau^\Pi \geq V_\tau^* - \epsilon$ for at least a fraction $1 - \delta$ of tasks.*

Clearly, an $(\epsilon, 1 - \delta)$ cover need not exist for an arbitrary committee $\Pi$ (if the committee is too small to cover enough tasks sufficiently well). There are, however, three knobs that we can adjust: $K$, $\epsilon$, and $\delta$. Next, we fix $\epsilon$ as exogenous, treating it effectively as a domain-specific hyperparameter, and suppose that $K$ is a pre-specified bound on the maximum size of the committee.

**Problem 1.** *Fix the maximum committee size $K$ and $\epsilon$. Our goal is to find $\Pi$ which is a $(\epsilon, 1 - \delta)$-cover for the smallest $\delta \in [0, 1]$.*

Next, we present algorithmic approaches for this problem. Subsequently, Section 4, as well as our experimental results, vindicate this choice of the objective.

## 3 ALGORITHMS FOR LEARNING A POLICY COMMITTEE

In this section, we present algorithmic approaches for computing policy committees $\Pi$ to solve Problems 1. We consider the special case of the problem in which the tasks have a *structure representation*. Specifically, we assume that each task can be represented using a parametric model $\psi_\theta(s, a)$, where the parameters $\theta \in \mathbb{R}^d$ comprise both of the parameters of the transition distribution $\mathcal{T}$ and reward function $r$. Often, parametric task representation is given or direct; in cases when tasks are non-parametric, such as the Meta-World (Yu et al., 2020b), we can often use approaches for task embedding, such as LLM-based task representations (see Section 3.4). Consequently, we identify tasks $\tau$ with their representation parameters $\theta$ throughout, and overload $\Gamma$ to mean the distribution over task parameters, i.e., $\theta \sim \Gamma$.

### 3.1 A HIGH-LEVEL ALGORITHMIC FRAMEWORK

Even conventional RL presents a practical challenge in complex problems, as learning is typically time consuming and requires extensive hyperparameter tuning. Consequently, a crucial consideration in algorithm design is to minimize the number of RL runs we need to obtain a policy committee. To this end, we propose the following high-level algorithmic framework in which we only need $K$ independent (and, thus, entirely parallelizable) RL runs. This framework involves three steps:

1. SAMPLE $n$ tasks i.i.d. from $\Gamma$, obtaining $T = \{\theta_1, \ldots, \theta_n\}$ (parameters of associated tasks $\{\tau_1, \ldots, \tau_n\}$). In MTRL settings, $T$ is given.

2. CLUSTER the task set $T$ into $K$ subsets, each with an associated representative $\theta_k$, and

3. TRAIN policies $\pi_k$ for each cluster $k$ represented by $\theta_k$.

As we shall see presently (and demonstrate experimentally in both Subsection 5.3 and Appendix G.2), conventional clustering approaches are not ideally suited for our problem. We thus propose several alternative approaches which yield theoretical guarantees on the quality of $\Pi$ under mild conditions if all tasks share the transition dynamics and only differ in reward function. Empirically, we show that the proposed framework outperforms state of the art even when tasks also have distinct transition distributions.

### 3.2 CLUSTERING

The key aspect of our algorithmic design is clustering. We begin by providing a formal connection between the clustering step (step (2) of the framework above) and efficacy of optimal policies learned for each cluster (step (3) of the framework) using a variant of the simulation lemma (Lobel & Parr, 2024). This, in turn, provides us with a clustering objective that would yield formal guarantees about the efficacy of the policy committee we thereby obtain. For this result, we assume that each task has a shared dynamics, and a parametric reward function $r_\theta(s, a)$ where $\theta$ identifies a task-specific reward. While this is a theoretical limitation, we note that our subsequent clustering and training algorithms do not in themselves require this assumption, and our experimental results demonstrate that the overall approach is effective generally.

Let $\pi_i^*$ denote the optimal policy for task $\pi_i$. We use $V_i^{\pi_j^*}$ to denote the value of task $\tau_i$ using a policy that is optimal for task $\tau_j$.

**Lemma 1.** *Suppose that $r_\theta(s, a)$ is L-Lipschitz in $L_\infty$ norm, that is, for all $\theta, \theta'$, $\sup_{s,a} |r_\theta(s, a) - r_{\theta'}(s, a)| \leq L\|\theta - \theta'\|_\infty$. Then, for any two tasks $\tau_i$ and $\tau_j$ with respective $\theta_i$ and $\theta_j$ that satisfy $\|\theta_i - \theta_j\|_\infty \leq \epsilon$, $V_i^{\pi_j^*} \geq V_i^{\pi_i^*} - 2L\frac{1-\gamma^{h+1}}{1-\gamma}\epsilon$ if $\gamma < 1$ and $V_i^{\pi_j^*} \geq V_i^{\pi_i^*} - 2Lh\epsilon$ if $\gamma = 1$.*

Lipschitz continuity is a mild assumption; for example, it is satisfied by ReLU neural networks.

Next, we connect this to our ultimate goal as expressed in Problem 1.

**Definition 2.** *A set of representatives $C = \{\theta_1, \ldots, \theta_K\}$ is an $(\epsilon, 1 - \delta)$-parameter-cover for a task distribution $\Gamma$ if $\min_{\theta' \subset C} \|\theta - \theta'\|_\infty \leq \epsilon$ with probability at least $1 - \delta$ with respect to $\theta \sim \Gamma$.*

The following result then follows directly from Lemma 1.

**Theorem 2.** *Suppose $C$ is an $(\epsilon, 1 - \delta)$-parameter-cover for $\Gamma$ and $r_\theta(s, a)$ is L-Lipschitz in $L_\infty$, and let $\Pi$ contain a set of optimal policies to each $\theta \in C$. Then $\Pi$ is a $(2L\frac{1-\gamma^{h+1}}{1-\gamma}\epsilon, 1 - \delta)$-cover for $\Gamma$ when $\gamma < 1$ and $(2Lh\epsilon, 1 - \delta)$-cover when $\gamma = 1$.*[1]

This result enables us to focus on obtaining $(\epsilon, 1-\delta)$-parameter-cover guarantees *solely in the space of policy parameters*, at least when policies all share dynamics and differ only in reward functions. In particular, we consider the following clustering counterpart to the original problem:

**Problem 2.** *Fix $K$ and $\epsilon$. Our goal is to find $C$ with $|C| \leq K$ which is a $(\epsilon, 1-\delta)$-parameter-cover for the smallest $\delta \in [0, 1]$.*

Notably, while conventional clustering techniques, such as k-means, can be viewed as proxies for these objectives, there are clear differences insofar as the typical goal is to minimize sum of shortest distances of all vectors from cluster representatives, whereas our goal, essentially, is to "cover" as many vectors as we can. In Appendix G.2, we provide a histogram to illustrate the difference. Indeed, we show next that our problems are strongly inapproximable, *even if we restrict attention to $K = 1$.*

**Definition 3** (MAX-1-COVER). *Let $T = \{\theta_1, \ldots, \theta_n\} \subseteq \mathbb{R}^d$. Find $\theta \in \mathbb{R}^d$ which maximizes the size of $S \subseteq T$ with $\max_{\theta' \in S} \|\theta - \theta'\|_\infty \leq \epsilon$.*

**Theorem 3.** *For any $\epsilon > 0$ MAX-1-COVER does not admit an $n^{1-\epsilon}$ -approximation unless P = NP.*

We prove this in Appendix B via an approximation-preserving reduction from the Maximum Clique problem (Engebretsen & Holmerin, 2000).

Despite this strong negative result, we next design two effective algorithmic approaches. The first method runs in polynomial time and provides a constant-factor approximation, but it requires the dimension $d$ to be constant. The second is a general gradient-based approach.

**Greedy Elimination Algorithm** Before we discuss our main algorithmic approaches, we begin with an approach that provides a useful building block, but not theoretical guarantees. Consider a set $T$ of task parameter vectors, fix $K$, and suppose we wish to identify an $(\epsilon, 1 - \delta)$-parameter-cover with the smallest $\delta$ (Problem 2), but restrict attention to $\theta \in T$ in constructing such a cover. This problem is an instance of a MAX-K-COVER problem (where subsets correspond to sets covered by each $\theta \in T$), and can be approximated using a greedy algorithm which iteratively adds one $\theta \in T$ to $C$ that maximizes the most uncovered vectors in $T$. Its fixed-$\delta$ variant, on the other hand, is a set cover problem if $\delta = 0$, and a similar greedy algorithm approximates the minimum-$K$ cover $C$ for any $\delta$. However, neither of these algorithms achieves a reasonable approximation guarantee (as we can anticipate from Theorem 3), although our experiments show that greedy elimination is nevertheless an effective heuristic. But, as we show next, we can do better.

**Greedy Intersection Algorithm**

The key intuition for our contributed algorithm is that for any $\theta$, a $\epsilon$-hypercube centered at $\theta$ characterizes all possible $\theta'$ that can cover $\theta$ in the sense of Definition 2. Thus, if any pair of $\epsilon$-hypercubes centered at $\theta$ and $\theta'$ intersects, any point at the intersection covers both.

---

[1] We note that this and other results also work for the FS-MT-MDP setting with a finite set of tasks $T$. We omit this from the results for easier readability.

To see it more clearly, we provide the following example in the one-dimensional setting:

Each cross represents a parameter we aim to cover, while each line segment indicates the possible locations of the $\epsilon$-close representative for that parameter. By selecting a point within the overlapping region of these intervals, we can effectively cover their parameters simultaneously .

This may lead to a naive idea of iteratively constructing an intersection tree for all $\theta \in T$. Unfortunately, the size of such a tree is exponential in $n$ in the worst case because we need to check the intersection between every subset of parameters. Instead, we propose a GREEDY INTERSECTION algorithm, which is polynomial in $n$ that gets around this issue. The first stage of the algorithm is to create an intersection tree for each dimension independently. For $s$-th dimension, we sort the datapoints' $s$-th coordinates in ascending order. We refer to the sorted coordinates as $x_1 < x_2 \cdots < x_n$, and create a list for each point $x_i$ to remember how many other points can be covered together with it with initialization being $[x_i]$ itself.

Starting from the second smallest datapoint $x_2$, we check if $x_2 - \epsilon \leq x_1 + \epsilon$, i.e. if $x_2 \leq x_1 + 2\epsilon$. Since $x_2 - \epsilon > x_1 - \epsilon$ due to our sorting, any point inside $[x_2 - \epsilon, x_1 + \epsilon]$ can cover both $x_1, x_2$. Therefore if this interval is valid, we add $x_1$ to the list $[x_2]$ to indicate the existence of a simultaneous coverage for $x_1, x_2$. In general, for $x_i$, we check if $x_i \leq x_j + 2\epsilon$ with a descending $j = i - 1$ to 1 or until the condition no longer holds. If the inequality is satisfied, we add $x_j$ to $x_i$'s list. Then since we have ordered the set, for every index $j'$ less than $j$, $x_i > x_j + 2\epsilon > x_{j'} + 2\epsilon$. The coverage for all the $x$ in $x_i$'s list would be the interval $[x_i - \epsilon, x_j + \epsilon]$, where $j$ is the smallest index in $x_i$'s list. There are $1 + 2 + \cdots + n - 1 = \mathcal{O}(n^2)$ comparisons in total. We form a set of these lists, and call it $\mathcal{A}_s$ for the $s$-th dimension. The figure above illustrates how the algorithm works to find out $\mathcal{A}_1 = \{[x_1], [x_1, x_2], [x_1, x_2, x_3], [x_4, x_5]\}$.

The second stage is to find a hypercube covering the most points, consisting of an axis from each dimension. Due to the geometry of the Euclidean space, we know that two points $\theta_1, \theta_2$ are within $\epsilon$ in $\ell_\infty$-distance iff they appear inside one's list together for each dimension. Therefore, in order to find the maximum coverage with one hypercube such that its center is within $\ell_\infty$-distance to the most points, we wonder which combination of lists, $l_1 \ldots l_d$ each from the sets $\mathcal{A}_1 \ldots \mathcal{A}_d$ produces an intersection of the maximum cardinality. In our example, we can conclude that $[x_1, x_2, x_3], [x_4, x_5]$ need to be covered separately by two points between the blue or red vertical lines. The full algorithm is provided in the Appendix C.

Next, we show that GREED INTERSECTION yields provable coverage guarantees. We defer the proofs to Appendix D. For these results we use GI($K$) to refer to the solution (set $C = \{\theta_1, \ldots, \theta_K\}$) returned GREEDY INTERSECTION algorithm.

**Theorem 4.** *Suppose $T$ contains $n \geq \frac{9 \log(5/\alpha)}{2\beta^2}$ i.i.d. samples from $\Gamma$. Let $1 - \delta^*(K, \epsilon)$ be the optimal $(\epsilon, 1 - \delta)$-parameter-coverage of $\Gamma$ achievable given $K$. Then with probability at least $1 - \alpha$, GI(K) is a $\left(\epsilon, (1 - \frac{1}{e})(1 - \delta^*(K, \epsilon) - K\beta)\right)$-parameter-cover of $\Gamma$.*

The key limitation of GREEDY INTERSECTION is that it is exponential in $d$, and thus requires the dimension to be constant. This is a reasonable assumption in some settings, such as low-dimensional control. However, in many other settings, both $n$ and $d$ can be large. Our next algorithm addresses this issue.

**Gradient-Based Coverage** Consider Problem 2. For a finite set $T$, we can formalize this as the following optimization problem:

$$\max_{\{\theta_1, \ldots, \theta_K\}} \sum_{\theta \in T} \mathbf{1}(\min_{k \in [K]} \|\theta_k - \theta\|_\infty \leq \epsilon), \tag{1}$$

where $\mathbf{1}(\cdot)$ is 1 whenever the condition is true and 0 otherwise. However, objective equation 1 is non-convex and discontinuous. To address this, we propose the following differentiable proxy:

$$\min_{\substack{\{\theta_1,\ldots,\theta_K\}; \\ \alpha \in \mathbb{R}^{nK}}} \sum_i \mathbf{ReLU}\left(\left\{\sum_{k=1}^{K} \sigma_k(\alpha_i)\|\theta_k - \theta_i\|_{\infty}\right\} - \epsilon\right), \quad (2)$$

where $\sigma(\cdot)$ is a softmax function. Next, we demonstrate that this is a principled proxy by showing that when full coverage of $T$ is possible, solutions of objectives equation 1 and equation 2 coincide.

**Theorem 5.** *Fix $K$ and suppose $\exists \theta \in \{\theta_1, \ldots, \theta_K\}$ such that $\|\theta - \theta_i\| \leq \epsilon$ for all $i$. Then the sets of optimal solutions to equation 1 and equation 2 are equivalent.*

Thus, we can use gradient-based methods with objective in equation 2 to approximate solutions to Problem 2. Because the objective is still non-convex, we can improve performance by initializing with the solution obtained using GREEDY ELIMINATION or GREEDY INTERSECTION when $d$ is low.

### 3.3 TRAINING

The output of the CLUSTERING step above is a set of representative task parameters $C = \{\theta_1, \ldots, \theta_K\}$. The simplest way to use these to obtain a policy committee $\Pi$ is to train a policy $\pi_k$ optimized for each $\theta_k \in C$. However, this ignores the set of tasks that comprise each cluster $k$ associated with a representative $\theta_k$ (i.e., the set of tasks closest to $\theta_k$). As demonstrated empirically in the multi-task RL literature, using multiple tasks to learn a shared representation facilitates generalization (effectively enabling the model to learn features that are beneficial to all tasks in the cluster) (Sodhani et al., 2021; Sun et al., 2022; Yang et al., 2020b).

To address this, we propose an alternative which trains a policy $\pi_k$ to maximize the sum of rewards of the tasks in cluster $k$. Notably, our approach can use *any* RL algorithm to learn a policy associated with a cluster of tasks; in the experiments below, we use the most effective MTRL or meta-RL baseline for this purpose.

### 3.4 DEALING WITH NON-PARAMETRIC TASKS

Our approach assumes that tasks are parametric, so that we can reason (particularly in the clustering step) about parameter similarity. Many practical multi-task settings, however, are non-parametric, so that our algorithmic framework cannot be applied directly. In such cases, our approach can make use of any available method for extracting a parametric representation of an arbitrary task $\tau$. For example, it is often the case that tasks can be either described in natural language. We propose to leverage this property and use text embedding (e.g., from pretrained LLMs) as the parametric representation of otherwise non-parametric tasks, where this is feasible. Our hypothesis is that this embedding captures the most relevant semantic aspects of many tasks in practice, a hypothesis that our results below validate in the context of the Meta-World benchmark. This is analogous to what was done by Bing et al. (2023), with the main difference being that our task descriptions are with respect to higher-level goals, whereas Bing et al. (2023) describe tasks in terms of associated plans. We provide the full list of task descriptions for the Meta-World environment in Appendix H.

## 4 FEW-SHOT ADAPTATION

One application of learning a policy committee $\Pi$ that is a $(\epsilon, 1 - \delta)$-cover is that we can leverage it in meta-learning for few-shot adaptation. In particular, suppose that $\gamma = 1$. We now show that this translates into a few-shot sample complexity on a previously unseen task $\tau$ that is linear in $K$ (the size of the committee).

**Definition 4.** *The average expected reward for a given policy $\pi$ is measured per time step as*

$$\mu^{\pi}(s) = \lim_{h \to \infty} \frac{1}{h} \mathbb{E}\left[\sum_{t=1}^{h} r(s_t, \pi(s_t)) \mid s_0 = s\right].$$

*Define the empirical average reward of $\pi$ over $p$ episodes as $\hat{\mu}^\pi = \frac{1}{ph} \sum_{i=0}^{p} \sum_{t=0}^{h} r_t$. The bias of policy $\pi$ in state $s$ is defined as $\lambda^\pi(s) = \mathbb{E}[r(s, \pi(s)) + \lambda^\pi(s')] - \mu^\pi(s)$, and the span of the bias function is $\mathrm{sp}(\lambda^\pi) = \max_s \lambda^\pi(s) - \min_s \lambda^\pi(s)$.*

**Assumption 1.** *Suppose that each $\pi \in \Pi$ induces on the MDP $\mathcal{M}$ a single recurrent class with some additional transient states, i.e., $\mu^\pi(s) = \mu^\pi$ for all $s \in \mathcal{S}$, and $\mathrm{sp}(\lambda^\pi) \leq H$ for some finite $H$.*

The algorithmic idea is straightforward: evaluate each of $K$ policies in $\Pi$ by computing a sample average sum of rewards over $N$ randomly initialized episodes, and choose the best policy $\pi \in \Pi$ in terms of empirical average reward. This yields the following sample complexity bound.

**Theorem 6.** *Suppose $\Pi$ is a $(\epsilon, 1 - \delta)$-cover for $\Gamma$ and let $\tau \sim \Gamma$. Then if we run at least $p \geq \frac{32h(H+1)^2 \log(4/\alpha)}{(\beta - 2H)^2}$ episodes for each policy $\pi \in \Pi$, the policy maximizing $\hat{\mu}^\pi$ achieves $V_\tau^\pi \geq V_\tau^* - \epsilon - \beta$ with probability at least $1 - \delta - \alpha$, where $V_\tau^*$ is the optimal reward for $\tau$.*

## 5 Experiments

We study the effectiveness of our approach—PACMAN—in two environments, *MuJoCo* (Todorov et al., 2012) and *Meta-World* (Yu et al., 2020b). In the former, the tasks are low-dimensional and parametric, and we only vary the reward functions, whereas the latter has non-parametric robotic manipulation tasks with varying reward and transition dynamics.

**MuJoCo**  We selected two commonly used MuJoCo environments. The first is HalfCheetahVel where the agent has to run at different velocities, and rewards are based on the distance to a target velocity. The second is HumanoidDir where the agent has to move along the preferred direction, and the reward is the distance to the target direction. In both, we generate diverse rewards by randomly generating target velocity and direction, respectively, and use 100 tasks for training and another 100 for testing (in both zero-shot and few-shot settings), with parameters generated from a Gaussian mixture model with 5 Gaussians. In few-shot cases, we draw a single task for fine-tuning, and average the result over 10 tasks. For clustering, we use $K = 3, \epsilon = .6$, and use the gradient-based approach initialized with the result of the *Greedy Intersection* algorithm. For few-shot learning, we fine-tune all methods for 100 epochs.

**Meta-World**  It is a well-known multitask and meta-learning benchmark (Yu et al., 2020b). We focus on the set of robotic manipulation tasks in MT50, of which we use *30 for training and 20 for testing*. This makes the learning problem significantly more challenging than typical in prior MTRL and meta-RL work, where training sets are much larger compared to test sets (5 tests and 40 trains in the traditional MT45 setting). We leverage an LLM to generate a parameterization (Section 3.4) of the task. Specifically, text descriptions are fed to "Phi-3 Mini-128k Instruct" (Microsoft, 2024) and we compute the channel-wise mean over the features of penultimate layer as a 50 dimensional parameterization for each task. We use $K = 3$ and $\epsilon = .7$, which allows us to obtain an $(\epsilon, 1)$-parameter-cover for the set of training tasks in terms of $\ell_\infty$ norm with respect to the LLM-based task embedding.

### 5.1 Baselines and Evaluation

We compare our approaches to 10 state-of-the-art baselines. Five of these are designed for MTRL: 1) CMTA (Lan et al., 2024), 2) MOORE Hendawy et al. (2023), 3) CARE (Sodhani et al., 2021), 4) soft modulation (Soft) (Yang et al., 2020a), and 5) Multi task SAC (Yu et al., 2020b). Four more are meta-RL algorithms: 1) MAML (Finn et al., 2017a), 2) RL2 (Duan et al., 2017), 3) PEARL (Rakelly et al., 2019), 4) VariBAD (Zintgraf et al., 2020), and AMAGO (Grigsby et al., 2024). Finally, we also compare to the personalized RL approach using EM to learn a policy committee (EM) (Ivanov & Ben-Porat, 2024).

Our evaluation involves three settings: *training*, *zero-shot test*, and *few-shot test*. The training evaluation corresponds to standard MTRL. The zero-shot test evaluation uses the test set to evaluate all approaches but with no fine-tuning. Finally, our few-shot test evaluation allows a short round of fine-tuning on the test data. For PACMAN we select the best-performing policy for training and zero-shot test and use the proposed few-shot approach to *learn* the best policy through empirical policy evaluation for the few-shot test setting (see Section 4).

## 5.2 RESULTS

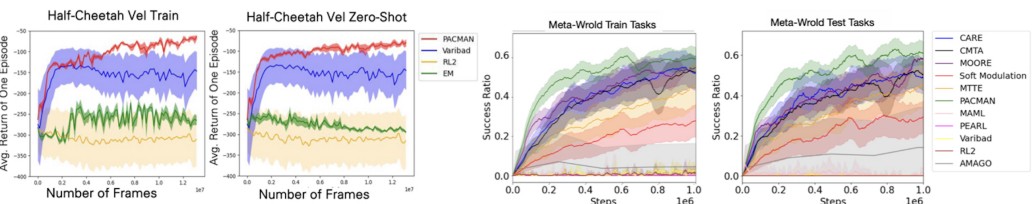

Figure 1: Left: Training and zero-shot test for HalfcheetahVel. Red, blue, green, yellow curves stand for PACMAN, VariBAD, EM, and RL2. Right: Training and zero-shot test for Meta-World.

**MuJoCo** In the MuJoCo environment, we focus on *personalization*, varying only reward functions and focusing on the ability to generalize to a diverse set of rewards. Consequently, our baselines here include meta-RL approaches (RL2, VariBAD) and EM (personalized RL, which requires the dynamics to be shared across tasks), and PACMAN uses VariBAD as the within-cluster RL method.

The first two plots in Figure 1 presents the MuJoCo results for the training and zero-shot test evaluations in HalfCheetahVel environment. We can see that PACMAN consistently and significantly outperforms the baselines, in both evaluations, with VariBAD the only competitive baseline. The advantage of PACMAN is also pronounced in HumanoidDir, whose results are deferred to G.

What is of particular interest is the few-shot comparison, which is provided in Table 1, where the advantage of PACMAN is especially notable. In Halfcheetah, the improvement over the best baseline (VariBAD) is by a factor of more than 2.5, while in Humanoid it is over 22%. From this, we can see the significant value of the PACMAN committee learning approach for few-shot adaptation.

Table 1: Few-shot learning effectiveness (MuJoCo).

|  | **Halfcheetah** | **Humanoid** |
|---|---|---|
| RL2 | -314.37 ± 1.15 | 946.17 ± 0.73 |
| VariBAD | -137.99 ± 1.14 | 1706.38 ± 0.75 |
| EM | -325.29 ± 1.84 | 947.06 ± 0.84 |
| **PACMAN** | **-54.03 ± 1.34** | **2086.50 ± 0.89** |

Table 2: Meta-World performance on 30 in-sample training tasks (left) and 20 out-of-sample test tasks (right). Performance is a moving average success rate for the last 2000 evaluation episodes over 3 seeds. Error bound is 1 sample standard deviation.

| | Train | | | Test (zero-shot) | | |
|---|---|---|---|---|---|---|
| **Method** | **500K Steps** | **1M Steps** | **Method** | **500K Steps** | **1M Steps** |
| Soft | 0.20 ± 0.08 | 0.28 ± 0.08 | Soft | 0.24 ± 0.10 | 0.29 ± 0.08 |
| MTTE | 0.29 ± 0.09 | 0.46 ± 0.11 | MTTE | 0.30 ± 0.09 | 0.45 ± 0.11 |
| CARE | 0.43 ± 0.08 | 0.52 ± 0.09 | CARE | 0.43 ± 0.08 | 0.49 ± 0.08 |
| CMTA | 0.43 ± 0.09 | 0.53 ± 0.08 | CMTA | 0.40 ± 0.08 | 0.51 ± 0.07 |
| MOORE | 0.44 ± 0.06 | 0.55 ± 0.01 | MOORE | 0.42 ± 0.07 | 0.58 ± 0.00 |
| **PACMAN** | **0.55 ± 0.04** | **0.60 ± 0.05** | **PACMAN** | **0.53 ± 0.05** | **0.61 ± 0.05** |

**Meta-World** Next, we turn to the complex multi-task Meta-World environment. In this environment, our approach uses MOORE for within-cluster training. Table 2 presents the results for training (standard MTRL setting) and zero-shot test evaluations, where we compare to the MTRL baselines (all meta-RL baselines are significantly worse on these metrics, likely because the goals of these algorithms are primarily efficacy in few-shot settings).

We observe that PACMAN again significantly outperforms all baselines after 500K training steps in both train and test cases (~25% improvement over the best baseline), though the gap is bridged somewhat after 1M steps. This shows that PACMAN trains considerably faster in this setting.

Table 3: Few-Shot Learning Results.

| **Method** | **6K Updates** | **12K Updates** |
|---|---|---|
| MAML | 0.0025 ± 0.006 | 0.01 ± 0.03 |
| PEARL | 0.03 ± 0.03 | 0.27 ± 0.07 |
| RL2 | 0.007 ± 0.01 | 0.02 ± 0.02 |
| VariBAD | 0.025 ± 0.06 | 0.027 ± 0.07 |
| AMAGO | 0.08 ± 0.09 | .093 ± 0.09 |
| Soft | 0.27 ± 0.07 | 0.26 ± 0.08 |
| MTTE | 0.37 ± 0.08 | 0.40 ± 0.10 |
| CARE | 0.39 ± 0.05 | 0.40 ± 0.06 |
| CMTA | 0.45 ± 0.07 | 0.34 ± 0.08 |
| MOORE | 0.41 ± 0.08 | 0.44 ± 0.11 |
| **PACMAN** | **0.53 ± 0.02** | **0.60 ± 0.02** |

Considering next the few-shot learning problem, the advantage of PACMAN over both MTRL and meta-RL baselines is particularly notable. The results are provided in Table 3. Performance is a moving average success rate for the last 2000 evaluation episodes over 3 seeds.

Error bound is 1 sample standard deviation. First, somewhat surprisingly, the meta-RL baselines, with the exception of PEARL, underperform MTRL baselines in this setting. This is because our evaluation is significantly more challenging, with only 30 training tasks but with 20 diverse test tasks, and the adaptation phase has a very short (6-12K updates) time horizon for few-shot training, than typical in prior work. In contrast, MTRL methods fare reasonably well. The proposed PACMAN approach, however, significantly outperforms all the baselines. For example, only 12K updates suffice to reliably identify the best policy (comparing with zero-shot results in Table 3), with the result outperforming the best baseline by $>36\%$.

## 5.3 Further Empirical Investigation of Our Algorithm

We investigate our algorithmic contribution in two ways. First we compare the difference between our method with both the popular clustering methods and the random clustering. Then we show how changing the number of policies in the committee influences its performance.

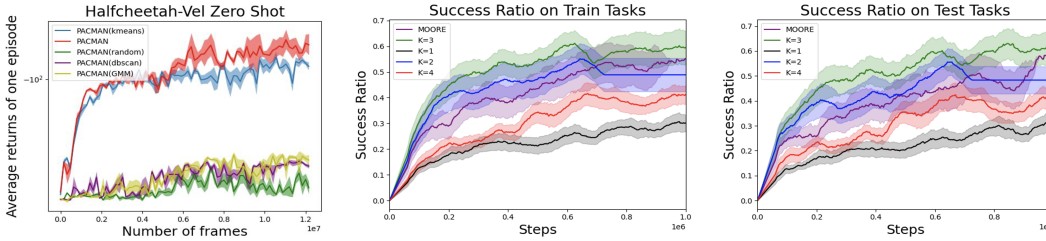

Figure 2: From left to right: (a) PACMAN ablation with different clustering methods ($K = 1, 2, 3, 4$; MuJoCo), (b) and (c) varying $K$ (training and zero-shot test, respectively, Meta-World).

First, Figure 2(a) shows that our clustering method indeed has the best performance. We emphasis that our improvement compared to KMeans is non-trivial, and a more detailed explanation is provided in G.2. Second, Table 4 demonstrates a clear advantage of utilizing a policy committee. Here, in few-shot settings, even using $K = 2$ already results in considerable improvement over the best baseline (MOORE), with $K = 3$ a significant further

Table 4: Few-shot in Meta-World, varying $K$.

| Method | 6K Updates | 12K Updates |
|---|---|---|
| MOORE | $0.42 \pm 0.06$ | $0.43 \pm 0.05$ |
| PACMAN ($K = 1$) | $0.32 \pm 0.05$ | $0.31 \pm 0.04$ |
| PACMAN ($K = 2$) | $0.50 \pm 0.05$ | $0.50 \pm 0.05$ |
| PACMAN ($K = 3$) | $0.61 \pm 0.04$ | $0.62 \pm 0.05$ |
| PACMAN ($K = 4$) | $0.32 \pm 0.05$ | $0.35 \pm 0.05$ |

boost. Another thing to note is that increasing $K$ is not always better. The results in both Figure 2(b) and (c), and Table 4 show that as the number of tasks becomes increasingly partitioned, the generalization ability of each committee member may weaken. Hence the performance for $K = 4$ is worse than $K = 3$. We also conducted the same experiments for Mujoco, the details are in G.2.

## 6 Conclusion and Limitations

We developed a general algorithmic framework for learning policy committees for effective generalization and few-shot learning in multi-task settings with diverse tasks that may be unknown at training time. We showed that our approach is theoretically grounded, and outperforms MTRL, meta-RL, and personalized RL baselines in both training, and zero-shot and few-shot test evaluations, often by a large margin. Nevertheless, our approach exhibits several important limitations. First, it requires tasks to be parametric, and while we demonstrate how LLMs can be used to effectively obtain task embeddings in the Meta-World environments, it is not clear how to do so generally. Second, it includes a scalar hyperparameter, $\epsilon$, which determines how we evaluate the quality of task coverage and needs to be adjusted separately for each environment, although this hyperparameter is easily tunable in practice.

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

APPENDIX

## A    PROOF OF LEMMA 1

$$
\begin{aligned}
V_i^{\pi_i^*} =& \mathbb{E}[\sum_{t=0}^{T} \gamma^t r_{\theta_i}(s_t, a_t) \,|\pi_i^*] = \mathbb{E}[\sum_{t=0}^{T} \gamma^t (r_{\theta_i}(s_t, a_t) - r_{\theta_j}(s_t, a_t) + r_{\theta_j}(s_t, a_t)) \,|\pi_i^*] \\
=& \mathbb{E}[\sum_{t=0}^{T} \gamma^t (r_{\theta_i}(s_t, a_t) - r_{\theta_j}(s_t, a_t)) \,|\pi_i^*] + \mathbb{E}[\sum_{t=0}^{T} \gamma^t r_{\theta_j}(s_t, a_t) \,|\pi_i^*] \\
=& \mathbb{E}[\sum_{t=0}^{T} \gamma^t (r_{\theta_i}(s_t, a_t) - r_{\theta_j}(s_t, a_t)) \,|\pi_i^*] + V_j^{\pi_i^*} \\
\leq& \sum_{t=0}^{T} \gamma^t L ||\theta_i - \theta_j||_\infty + V_j^{\pi_j^*}(-V_i^{\pi_j^*} + V_i^{\pi_j^*}) \\
\leq& L \frac{\gamma^{T+1}-1}{\gamma-1}\epsilon + (V_2^{\pi_j^*} - V_i^{\pi_j^*}) + V_i^{\pi_j^*} = L\frac{\gamma^{T+1}-1}{\gamma-1}\epsilon + \mathbb{E}[\sum_{t=0}^{T}\gamma^t r_{\theta_i}(s_t,a_t) - r_{\theta_j}(s_t,a_t)\,|\pi_j^*] + V_i^{\pi_j^*} \\
\leq& 2L\frac{\gamma^{T+1}-1}{\gamma-1}\epsilon + V_i^{\pi_j^*}
\end{aligned}
$$

If the discount factor $\gamma = 1$, the argument is as follows:

$$
\begin{aligned}
V_i^{\pi_i^*} =& \mathbb{E}[\sum_{t=0}^{T} r_\theta(s_t, a_t) \,|\pi_i^*] = \mathbb{E}[\sum_{t=0}^{T} r_\theta(s_t, a_t) - r_{\theta_j}(s_t, a_t) + r_{\theta'}(s_t, a_t) \,|\pi_i^*] \\
=& \mathbb{E}[\sum_{t=0}^{T} r_\theta(s_t, a_t) - r_{\theta_j}(s_t, a_t) \,|\pi_i^*] + \mathbb{E}[\sum_{t=0}^{T} (r_{\theta_j}(s_t, a_t) \,|\pi_i^*] \\
=& \mathbb{E}[\sum_{t=0}^{T} r_{\theta_i}(s_t, a_t) - r_{\theta_j}(s_t, a_t) \,|\pi_i^*] + V_j^{\pi_i^*} \\
\leq& \sum_{t=0}^{T} L||\theta_i - \theta_j||_\infty + V_j^{\pi_j^*}(-V_i^{\pi_j^*} + V_i^{\pi_j^*}) \\
\leq& TL\epsilon + (V_j^{\pi_j^*} - V_i^{\pi_j^*}) + V_i^{\pi_j^*} = TL\epsilon + \mathbb{E}[\sum_{t=0}^{T} r_{\theta_i}(s_t,a_t) - r_{\theta_j}(s_t,a_t)\,|\pi_j^*] + V_i^{\pi_j^*} \\
\leq& 2TL\epsilon + V_i^{\pi_j^*}
\end{aligned}
$$

## B    PROOF OF THEOREM 3

**Definition 5** (Gap preserving reduction for a maximization problem). *Assume $\Pi_1$ and $\Pi_2$ are some maximization problems. A gap-preserving reduction from $\Pi_1$ to $\Pi_2$ comes with four parameters (functions) $f_1, \alpha, f_2$ and $\beta$. Given an instance $x$ of $\Pi_1$, the reduction computes in polynomial time an instance $y$ of $\Pi_2$ such that: $OPT(x) \geq f_1(x) \implies OPT(y) \geq f_2(y)$ and $OPT(x) < \alpha|x|f_1(x) \implies OPT(y) < \beta|y|f_2(y)$.*

*Proof.* Let $G = (V, E)$ be an undirected graph with n vertices and m edges. We create an instance of Max-coverage for a set of $\theta$s in $\mathbb{R}^n$ by filling out their coordinate matrix $A_{ij} =$

$$
\begin{cases}
0 & \text{if } i = j \\
1.5\epsilon & \text{if } i, j \text{ are adjacent} \\
2.5\epsilon & \text{if } i, j \text{ are not adjacent}
\end{cases}
$$

For example in the graph below $x_3, x_4$ are $x_5$'s neighbors, but $x_1, x_2$ are not.

| dim | $\theta_1$ | $\theta_2$ | $\theta_3$ | $\theta_4$ | $\theta_5$ |
|---|---|---|---|---|---|
| 1 | 0 | 2.5 | 2.5 | 2.5 | 2.5 |
| 2 | 2.5 | 0 | 2.5 | 2.5 | 2.5 |
| 3 | 2.5 | 2.5 | 0 | 2.5 | 1.5 |
| 4 | 2.5 | 2.5 | 2.5 | 0 | 1.5 |
| 5 | 2.5 | 2.5 | 1.5 | 1.5 | 0 |

Let $\theta_1 = [0, 2.5, 2.5, 2.5, 2.5], \theta_2 = [2.5, 0, 2.5, 2.5, 2.5], \theta_3 = [2.5, 2.5, 0, 2.5, 1.5], \theta_4 = [2.5, 2.5, 2.5, 0, 2.5], \theta_5 = [2.5, 2.5, 1.5, 1.5, 0]$.

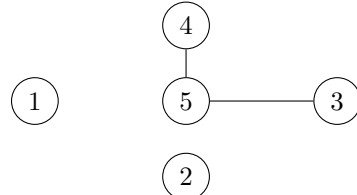

Projected onto the fifth axis, our thetas look like:

$$x_5 - \epsilon \xrightarrow{\quad\times\quad} x_5 + \epsilon$$
$$x_1 - \epsilon \xrightarrow{\quad\times\quad} x_1 + \epsilon$$
$$x_2 - \epsilon \xrightarrow{\quad\times\quad} x_2 + \epsilon$$
$$x_3 - \epsilon \xrightarrow{\quad\times\quad} x_3 + \epsilon$$
$$x_4 - \epsilon \xrightarrow{\quad\times\quad} x_4 + \epsilon$$

And similarly, onto the third axis:

$$x_3 - \epsilon \xrightarrow{\quad\times\quad} x_3 + \epsilon$$
$$x_1 - \epsilon \xrightarrow{\quad\times\quad} x_1 + \epsilon$$
$$x_2 - \epsilon \xrightarrow{\quad\times\quad} x_2 + \epsilon$$
$$x_4 - \epsilon \xrightarrow{\quad\times\quad} x_4 + \epsilon$$
$$x_5 - \epsilon \xrightarrow{\quad\times\quad} x_5 + \epsilon$$

We claim that we have constructed a gap-preserving reduction for any $t > 0$

$$OPT(A) = n \implies OPT(B) = n$$

$$OPT(A) < n^{1-t} \implies OPT(B) < n^{1-t}.$$

To begin with, if the Max-Clique instance consists of a complete graph, then the $\theta$s we created have coordinates equal to $1.5\epsilon$ everywhere except $i$-th coordinate, which is zero. So they can all be covered by one $\tilde{\theta} = [0.7\epsilon, 0.7\epsilon, \ldots, 0.7\epsilon]$, the coverage size is $n$. Therefore, the first implication is true.

Then for the second statement, we argue with the contrapositive: assume that one of the maximum coverage sets is $S = \{i_1, \ldots, i_k\}$ and $k \geq n^{1-t}$. We have to prove that the maximum clique has size greater than or equal to $k \geq n^{1-t}$.

Specifically, we prove that the vertices corresponding to the elements from $S$ form a clique.

If $\theta_i, \theta_j$ are from the set $S$, then they should be covered on each dimension since the $||\theta_i - \theta_j||_\infty = \max |\theta_i^d - \theta_j^d| \leq \epsilon$. So $\theta_i, \theta_j$ have to be adjacent, because otherwise their corresponding coordinates on the $i$-th and $j$-th dimension are more than $\epsilon$ away. For example, we have theta $\theta_3^3 = 0$ and $\theta_5^5 = 0$, so $\theta_5^3$ and $\theta_5^3$ must be $1.5\epsilon$ rather than $2.5\epsilon$, which indicates that $3, 5$ are neighbors in the graph.

Therefore, the points in $S$ correspond to a clique of size $k \geq n^{1-t}$ in the graph. Thus, if the graph $G$ has a clique of size less than $n^{1-t}$, then the maximum coverage set has size less than $n^{1-t}$. $\square$

## C  PSEUDOCODE OF GREEDY INTERSECTION

The full pseudocode for the *Greedy Intersection* algorithm is provided as Algorithm 1.

---

**Algorithm 1** Greedy Intersection

---

**Input**: $T = \{\theta_i\}_{i=1}^N, \epsilon > 0, K \geq 1$
**Output**: Parameter cover $C$

1:  $C \leftarrow []$
2:  **for** $round\ k = 1$ to $K$ **do**
3:      **for** $dimension\ m = 1$ to $d$ **do**
4:          Sort $T$ in ascending order based on their $m$-th coordinates
5:          $lists_m \leftarrow []$
6:          **for** $indiviual\ i = 2$ to $N$ **do**
7:              $S_i \leftarrow [\theta_i]$
8:              **for** $j = i - 1$ to $1$ **do**
9:                  **if** $\theta_i$'s $m$-th coordinate $< \theta_j$'s $m$-th coordinate $+2\epsilon$ **then**
10:                     Add $\theta_j$ to $S_i$
11:                 **else**
12:                     **if** $lists_m[-1] \subseteq S_i$  **then**
13:                         $lists_m[-1] \leftarrow S_i$
14:                     **else**
15:                         Add $S_i$ to $lists_m$
16:                     **end if**
17:                     **break**
18:                 **end if**
19:             **end for**
20:         **end for**
21:      **end for**
22:      $S^{1*}, \ldots, S^{m*} \leftarrow \operatorname{argmax}_{S^1 \in lists_1, \ldots, S^m \in lists_m} |S^1 \cap \cdots \cap S^m|$
23:      $covered \leftarrow S^{1*} \cap \cdots \cap S^{m*}$
24:      $\hat{\theta}_k \leftarrow$ average of the $covered$
25:      $T \leftarrow T - covered$
26:      $C$.adds($\hat{\theta}_k$)
27: **end for**
28: **return** $C$

---

## D  PROOF OF THEOREM 4

Based on the proof of maxmizing monotone submodular functions by Nemhauser et al. (1978).

**Lemma 7.** *Suppose $1 - \delta^*(K)$ is the optimal $(\epsilon, 1 - \delta)$-parameter-cover of $\Gamma$ achievable with fixed $K$. With probability at least $1 - \alpha$, the probability of $\theta$ from $\Gamma$ getting covered by the first $i$ representatives generated by Algorithm 1 is greater than $\frac{1 - \delta^*(K) - K\beta}{K} \sum_{j=0}^{i-1}(1 - 1/K)^j$.*

*Proof.* We will prove the lemma through induction. We begin by defining the coverage region of each of the $K$ committee member in the optimal parameter-cover as $S_i^*$. Furthermore, let $\Pi^*$ denote the region covered by this optimal parameter-cover. Thus, $\Pi^* = \bigcup S_i^*$. Next, let $A_i$ denote the region covered by the representative selected on the $i$-th iteration. And let $C_i$ denote the set of $\theta$s from the dataset $T$ that are covered after $i$-th iteration.

First of all, we want to show at $i = 1$, the probability for $\theta \sim \Gamma$ getting covered is greater than $\frac{1 - \delta^*(K) - K\beta}{K} \sum_{j=0}^{0}(1 - 1/K)^0 = \frac{1 - \delta^*(K) - K\beta}{K}$.

By Hoeffding's theorem, $\Pr_{\theta \sim \Gamma}[\mathbb{E}_{\theta \sim \Gamma}[\mathbf{1}(\theta \in \bigcup_{j=1}^{i-1} A_j)] - \frac{\sum_i \mathbf{1}(\theta_i \in \bigcup_{j=1}^{i-1} A_j)}{N}) \geq \frac{\beta}{3}] \leq \exp(-2N\beta^2/9) = \frac{\alpha}{5}$. Hence, with probability at least $1 - \frac{\alpha}{5}$, $\Pr_{\theta \sim \Gamma}[\theta \in \bigcup_{j=1}^{i-1} A_j] = \mathbb{E}_{\theta \sim \Gamma}[\mathbf{1}(\theta \in \bigcup_{j=1}^{i-1} A_j)] \leq \frac{\sum_i \mathbf{1}(\theta_i \in \bigcup_{j=1}^{i-1} A_j)}{N} + \frac{\beta}{3} = \frac{|C_{i-1}|}{N} + \frac{\beta}{3}$.

Now the union bound first gives that $\Pr_{\theta\sim\Gamma}[\theta \in \Pi^* \wedge \theta \notin \bigcup_{j=1}^{i-1} A_j] \geq \Pr_{\theta\sim\Gamma}[\theta \in \Pi^*] - \Pr_{\theta\sim\Gamma}[\theta \in \bigcup_{j=1}^{i-1} A_j] = 1 - \delta^*(K) - \Pr_{\theta\sim\Gamma}[\theta \in \bigcup_{j=1}^{i-1} A_j]$. Applying union bound again, we obtain that with probability at least $1 - \alpha_1$, $\sum_{i=1}^{K} \Pr_{\theta\sim\Gamma}[\theta \in S_i^* \wedge \theta \notin \bigcup_{j=1}^{i-1} A_j] \geq \Pr_{\theta\sim\Gamma}[\theta \in \Pi^* \wedge \theta \notin \bigcup_{j=1}^{i-1} A_j] \geq 1 - \delta^*(K) - (\frac{|C_{i-1}|}{N} + \frac{\beta}{3})$. Hence, $\max_{i \in [K]} \Pr_{\theta\sim\Gamma}[\theta \in S_i^* \wedge \theta \notin \bigcup_{j=1}^{i-1} A_j] \geq \frac{1 - \delta^*(K) - (\frac{|C_{i-1}|}{N} + \frac{\beta}{3})}{K}$. Let us call this maximising $S_i^*$ $\hat{S}$.

According to our Algorithm 1, $A_i$ covers the most $\theta$s from $T$ that were not covered in the previous rounds by $\bigcup_{j=1}^{i-1} A_j$. In particular, $|C_i| - |C_{i-1}|$ is greater or equal to the number of $\theta$s from $T$ covered in $\hat{S}$ but not $\bigcup_{j=1}^{i-1} A_j$. Let us denote the latter as $s_1$, and the former as $s_2$, then $s_1 - s_2 \leq 0$.

Hoeffding's theorem gives us $\Pr_{\theta\sim\Gamma}(\mathbb{E}_{\theta\sim\Gamma}[\mathbf{1}[\theta \in \hat{S} \wedge \theta \notin \bigcup_{j=1}^{i-1} A_j]] - s_1/N) \geq \frac{\beta}{6}) \leq (\frac{\alpha}{5})^4$ and $\Pr_{\theta\sim\Gamma}(s_2/N - \mathbb{E}_{\theta\sim\Gamma}[\mathbf{1}[\theta \in A_i \wedge \theta \notin \bigcup_{j=1}^{i-1} A_j]] \geq \frac{\beta}{6}) \leq (\frac{\alpha}{5})^4$. Hence with probability at least $1 - 2(\frac{\alpha}{5})^4$, $\mathbb{E}_{\theta\sim\Gamma}[\mathbf{1}[\theta \in \hat{S} \wedge \theta \notin \bigcup_{j=1}^{i-1} A_j]] - \mathbb{E}_{\theta\sim\Gamma}[\mathbf{1}[\theta \in A_i \wedge \theta \notin \bigcup_{j=1}^{i-1} A_j]] = (\mathbb{E}_{\theta\sim\Gamma}[\mathbf{1}[\theta \in \hat{S} \wedge \theta \notin \bigcup_{j=1}^{i-1} A_j]] - s_1/N) + (s_1 - s_2)/N + (s_2/N - \mathbb{E}_{\theta\sim\Gamma}[\mathbf{1}[\theta \in A_i \wedge \theta \notin \bigcup_{j=1}^{i-1} A_j]] \leq \frac{\beta}{6} + \frac{\beta}{6} = \frac{\beta}{3}$.

Applying the result we obtained at the beginning of the proof, we have with probability at least $1 - \frac{\alpha}{5} - 2(\frac{\alpha}{5})^4$,

$$\Pr_{\theta\sim\Gamma}[\theta \in A_i \wedge \theta \notin \bigcup_{j=1}^{i-1} A_j] \geq \Pr_{\theta\sim\Gamma}[\theta \in \hat{S} \wedge \theta \notin \bigcup_{j=1}^{i-1} A_j] - \frac{\beta}{3} \geq \frac{1 - \delta^*(K) - (\frac{|C_{i-1}|}{N} + \frac{\beta}{3})}{K} - \frac{\beta}{3}.$$

(3)

Since nothing is covered before the first iteration, we can use equation 3 with $|C_0| = 0$ to prove the base condition for the claim. Because $K \geq 1$, we have $\frac{1 - \delta^*(K) - \frac{\beta}{3}}{K} - \frac{\beta}{3} = \frac{1 - \delta^*(K) - \frac{(1+K)\beta/3}{K}}{K} \geq \frac{1 - \delta^*(K) - K\beta}{K}$.

The induction hypothesis is that for all $i \leq K - 1$, we have $\Pr_{\theta\sim\Gamma}[\theta \in \bigcup_{j=1}^{i} A_j] \geq \frac{1 - \delta^*(K) - K\beta}{K} \sum_{j=0}^{i}(1 - 1/K)^j$.

By Hoeffding, $\Pr_{\theta\sim\Gamma}[|\Pr_{\theta\sim\Gamma}[\theta \in \bigcup_{j=1}^{i-1} A_j] - \frac{|C_{i-1}|}{N}| \geq \beta/3] \leq 2\exp(-2N\beta^2/9)$. In other words, with probability at least $1 - 2\frac{\alpha}{5}$, $\Pr_{\theta\sim\Gamma}[\theta \in \bigcup_{j=1}^{i-1} A_j] \geq \frac{|C_{i-1}|}{N} - \beta/3$ and $\frac{|C_{i-1}|}{N} \geq \Pr_{\theta\sim\Gamma}[\theta \in \bigcup_{j=1}^{i-1} A_j] - \beta/3$.

Then at the step $i = K$, since for $\frac{\alpha}{5} \in (0, 1), (\frac{\alpha}{5})^4 < \frac{\alpha}{5}$, we have with probability at least $1 - 2\frac{\alpha}{5} - \frac{\alpha}{5} - 2(\frac{\alpha}{5})^4 \geq 1 - 5\frac{\alpha}{5} = 1 - \alpha$,

$$\begin{aligned}
\Pr_{\theta\sim\Gamma}[\theta \in \bigcup_{j=1}^{i} A_j] &= \Pr_{\theta\sim\Gamma}[\theta \in \bigcup_{j=1}^{i-1} A_j] + \Pr_{\theta\sim\Gamma}[\theta \in A_i \wedge \theta \notin \bigcup_{j=1}^{i-1} A_j] \\
&\geq \frac{|C_{i-1}|}{N} - \frac{\beta}{3} + \frac{1 - \delta^*(K) - (\frac{|C_{i-1}|}{N} + \beta/3)}{K} - \frac{\beta}{3} \\
&= \frac{1 - \delta^*(K)}{K} + (1 - 1/K)\frac{|C_{i-1}|}{N} - \frac{(2K+1)\beta}{3K} \\
&\geq \frac{1 - \delta^*(K)}{K} + (1 - 1/K)(\Pr_{\theta\sim\Gamma}[\theta \in \bigcup_{j=1}^{i-1} A_j] - \beta/3) - \frac{(2K+1)\beta}{3K} \\
&\geq \frac{1 - \delta^*(K)}{K} + (1 - 1/K)(\frac{1 - \delta^*(K) - K\beta}{K} \sum_{j=0}^{i-1}(1 - 1/K)^j) - (1 - 1/K)\beta/3 - \frac{(2K+1)\beta}{3K} \\
&= \frac{1 - \delta^*(K)}{K} - \frac{(2K + 1 + K - 1)\beta}{3K} + \frac{1 - \delta^*(K) - K\beta}{K} \sum_{j=1}^{i}(1 - 1/K)^j
\end{aligned}$$

$$=\frac{1-\delta^*(K)-K\beta}{K}\sum_{j=0}^{i}(1-1/K)^j$$

$\square$

*Proof of Theorem 4.* We can directly apply lemma 7 to $i = K$. Call the region defined by the cover generated by Algorithm 1 $\Pi_K = \bigcup_{j=1}^{K} A_j$. Using the inequality $(1-1/K)^K \geq 1-1/e$ for all $K \geq 0$, we have

$$\Pr_{\theta\sim\Gamma}[\theta \in \Pi_K] \geq \frac{1-\delta^*(K)-K\beta}{K}\sum_{j=0}^{K}(1-1/K)^j = \frac{1-\delta^*(K)-K\beta}{K}\frac{1-(1-1/K)^K}{1-(1-1/K)}$$

$$=(1-\delta^*(K)-K\beta)(1-(1-1/K)^K) \geq (1-1/e)(1-\delta^*(K)-K\beta).$$

$\square$

# E  PROOF OF THEOREM 5

*Proof.* Let us call the optimal solutions set to equation 1 $A_1$, and the optimal solutions set to equation 2 $A_2$.

We first show $A_1 \subset A_2$. Pick any $\{\theta_1, \ldots, \theta_K\} \in A_1$. Due to the premise, for each $i$, since $\min_{k\in[K]}\|\theta_k - \theta_i\|_\infty - \epsilon \leq 0$, there exists $\theta_{k^*}$ such that $\|\theta_{k^*} - \theta_i\|_\infty - \epsilon \leq 0$. Thus, we can have $\alpha_{k^*}(i) = 1$, and $\alpha_k(i) = 0$ for all the other $k \neq k^*$. Then $\mathbf{ReLU}\left(\left\{\sum_{k\in[K]}\mathrm{softmax}_k(\alpha_i)\|\theta_k - \theta_i\|_\infty\right\} - \epsilon\right) = \mathbf{ReLU}(\|\theta_{k^*} - \theta_i\|_\infty - \epsilon) = 0$. By setting $\alpha$ this way, we could achieve the zero loss for the relaxation problem. Hence $\{\theta_1, \ldots, \theta_K\} \in A_2$.

Now to show $A_2 \subset A_1$, suppose $\{\theta_1, \ldots, \theta_K\}, \alpha$ is a optimal solution. Due to the premise, we must have that $\mathbf{ReLU}\left(\left\{\sum_{k\in[K]}\mathrm{softmax}_k(\alpha_i)\|\theta_k - \theta_i\|_\infty\right\} - \epsilon\right) = 0$ for each $i$. Now fix $i$, since $\mathrm{softmax}(\alpha)$ is nonnegative and sums to 1, there must be some positive coordinate $\mathrm{softmax}_{k'}(\alpha_i)$. Hence for all such $k'$, $\mathbf{ReLU}(\|\theta_{k'} - \theta_i\|_\infty - \epsilon) = 0$, i.e., $\|\theta_{k'} - \theta_i\|_\infty \leq \epsilon$. Thus, $\min_{k\in[K]}\|\theta_k - \theta_i\|_\infty \leq \|\theta_{k'} - \theta_i\|_\infty \leq \epsilon$ also holds, and $\{\theta_1, \ldots, \theta_K\}\sum_i \mathbf{1}(\min_{k\in[K]}\|\theta_k - \theta_i\|_\infty \leq \epsilon) = n$. Consequently, $\{\theta_1, \ldots, \theta_K\} \in A_1$. $\square$

# F  PROOF OF THEOREM 6

We prove this by leveraging the following lemma by Azar et al. (2013).

**Lemma 8.** *(Azar et al., 2013, Lemma 1) Under Assumption 1, $|\hat{\mu}^\pi - \mu^\pi| \leq 2(H+1)\sqrt{\frac{2\log(2/\alpha)}{ph}} + \frac{H}{h}$ with probability at least $1 - \alpha$.*

*Proof.* Let $p = \frac{32h(H+1)^2\log(4/\alpha)}{(\beta-2H)^2}$. Denote the average rewards of the best and second best policy in the committee as $\mu^+, \mu^-$. If $\mu^+ - \mu^- > \beta/h$, by ensuring the difference between the estimation and the true average reward is small than $\beta/2h$. We can make sure we have picked the best policy. From Lemma 8, we know $\Pr[\hat{\mu}^- \leq \mu^- + 2(H+1)\sqrt{\frac{2\log(4/\alpha)}{ph}} + \frac{H}{h}] = \Pr[\hat{\mu}^- \leq \mu^- + \beta/2h] \geq 1 - \alpha/2$. And $\Pr[\hat{\mu}^+ \geq \mu^+ - 2(H+1)\sqrt{\frac{2\log(4/\alpha)}{ph}} + \frac{H}{h}] = \Pr[\hat{\mu}^+ \leq \mu^+ - \beta/2h] \geq 1 - \alpha/2$. Hence with probability at least $1 - \alpha$, $\hat{\mu}^+ > \mu^+ - \beta/2h \geq \mu^- + \beta/h - \beta/2h = \mu^- + \beta/2h > \hat{\mu}^-$. Thus the empirically best policy we have picked is also the best in expectation. Now if $\mu^+ - \mu^- < \beta/h$, no matter which one we pick, we have the difference bound by $\beta/h$. The same holds for all pairs of policies ordered based on their expected values. Either way, with probability $1 - \alpha$, we could find the best policy in the committee. Since our committee is a $(\epsilon, 1 - \delta)$ cover, we are able to pick the policy with suboptimality $\beta + \epsilon$ with probability $1 - \delta - \alpha$. $\square$

# G ADDITIONAL EMPIRICAL RESULTS

## G.1 RESULTS ON HUMANOID DIRECTION

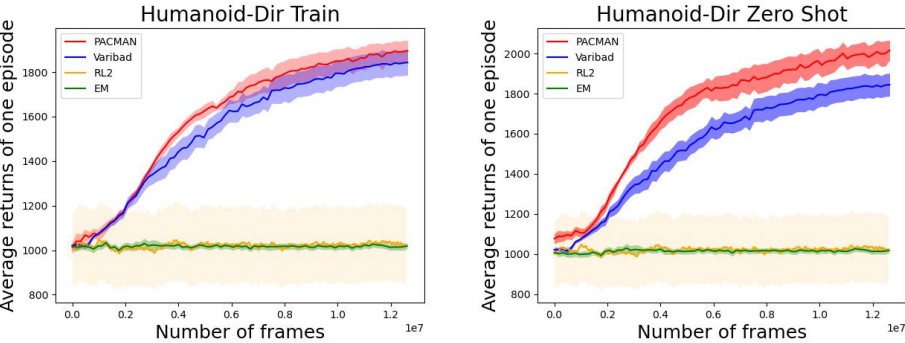

Figure 3: Left: Humanoid (direction) training. Right: Humanoid (direction) zero-shot.

Humanoid's few shot result has been listed in Table 1.

## G.2 ADDITIONAL RESULTS FOR EMPIRICAL INVESTIGATION OF OUR METHOD

### G.2.1 ABLATIONS OVER CLUSTERING METHODS

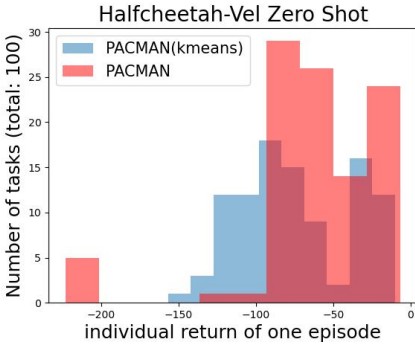

Figure 4: Histogram comparison of two clustering methods for zero-shot individual task rewards in Half-Cheetah (velocity).

While the performance of the KMeans algorithm appears relatively close to our method due to the significant gap between it and the other three clustering methods (DBSCAN, GMM, and Random), we emphasize that this result considers one hundred percent of the population.

The advantage of our algorithm becomes even more apparent when focusing on the welfare of the majority. To illustrate this, we present a histogram of rewards for individual test tasks during zero-shot testing using policies trained with our algorithm versus KMeans on the Half-Cheetah (velocity) benchmark.

The results vividly highlight a significantly greater density of high-performing tasks (red regions on the right) with our method. This suggests that our approach effectively promotes superior task performance while minimizing underperformance. In contrast, the KMeans method yields a more uniform but mediocre distribution of task performance. There is an ideological difference between these two clustering methods.

### G.2.2 HYPERPARAMETER ABLATIONS

We consider here additional ablations varying $K$ and $\epsilon$ omitted from the main body.

First, we present the results of ablations on K on Mujoco (Halfcheetah-Velocity).

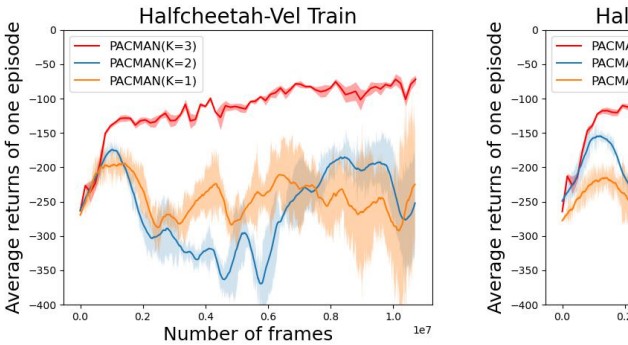

Figure 5: Varying $K$ from 1 to 3 for Halfcheetah-Velocity.

Next, we show the effect of the $\epsilon$ hyperparmater in the Meta-World zero shot setting. These results are reported for success rate across all tasks.

|  | $\epsilon = .4$ | $\epsilon = .7$ | $\epsilon = 1$ |
|---|---|---|---|
| 500K Steps | 0.05 | 0.28 | 0.29 |
| 1M Steps | 0.05 | 0.31 | 0.40 |

Table 5: Performance metrics for different $\epsilon$ values at 500K and 1M steps.

We find that increasing $\epsilon$ to cover more tasks can also improve performance (for a similar reason that increasing $K$ may not, as higher $\epsilon$ can ensure that we do not end up with clusters with too few tasks). Of course, for sufficiently high $\epsilon$, only a single cluster will emerge, so this, too induces an interesting tradeoff.

# H META-WORLD TASK DESCRIPTIONS

| Task Name | Objective | Environment Details |
|---|---|---|
| Reach-v1 | Move the robot's end-effector to a target position. | The task is set on a flat surface with random goal positions. The target position is marked by a small sphere or point in space. |
| Push-v1 | Push a puck to a specified goal position. | The puck starts in a random position on a flat surface. The goal position is marked on the surface. |
| Pick-Place-v1 | Pick up a puck and place it at a designated goal position. | The puck is placed randomly on the surface. The goal position is marked by a target area. |
| Door-Open-v1 | Open a door with a revolving joint. | The door can be opened by rotating it around the joint. Door positions are randomized. |
| Drawer-Open-v1 | Open a drawer by pulling it. | The drawer is initially closed and can slide out on rails. |
| Drawer-Close-v1 | Close an open drawer by pushing it. | The drawer starts in an open position. |
| Button-Press-Topdown-v1 | Press a button from the top. | The button is mounted on a panel or flat surface. |
| Peg-Insert-Side-v1 | Insert a peg into a hole from the side. | The peg and hole are aligned horizontally. |
| Window-Open-v1 | Slide a window open. | The window is set within a frame and can slide horizontally. |
| Window-Close-v1 | Slide a window closed. | The window starts in an open position. |
| Door-Close-v1 | Close a door with a revolving joint. | The door can be closed by rotating it around the joint. |
| Reach-Wall-v1 | Bypass a wall and reach a goal position. | The goal is positioned behind a wall. |
| Pick-Place-Wall-v1 | Pick a puck, bypass a wall, and place it at a goal position. | The puck and goal are positioned with a wall in between. |
| Push-Wall-v1 | Bypass a wall and push a puck to a goal position. | The puck and goal are positioned with a wall in between. |
| Button-Press-v1 | Press a button. | The button is mounted on a panel or surface. |
| Button-Press-Topdown-Wall-v1 | Bypass a wall and press a button from the top. | The button is positioned behind a wall on a panel. |
| Button-Press-Wall-v1 | Bypass a wall and press a button. | The button is positioned behind a wall. |
| Peg-Unplug-Side-v1 | Unplug a peg sideways. | The peg is inserted horizontally and needs to be unplugged. |
| Disassemble-v1 | Pick a nut out of a peg. | The nut is attached to a peg. |
| Hammer-v1 | Hammer a nail on the wall. | The robot must use a hammer to drive a nail into the wall. |
| Plate-Slide-v1 | Slide a plate from a cabinet. | The plate is located within a cabinet. |
| Plate-Slide-Side-v1 | Slide a plate from a cabinet sideways. | The plate is within a cabinet and must be removed sideways. |
| Plate-Slide-Back-v1 | Slide a plate into a cabinet. | The robot must place the plate back into a cabinet. |
| Plate-Slide-Back-Side-v1 | Slide a plate into a cabinet sideways. | The plate is positioned for a sideways entry into the cabinet. |
| Handle-Press-v1 | Press a handle down. | The handle is positioned above the robot's end-effector. |

| Handle-Pull-v1 | Pull a handle up. | The handle is positioned above the robot's end-effector. |
|---|---|---|
| Handle-Press-Side-v1 | Press a handle down sideways. | The handle is positioned for sideways pressing. |
| Handle-Pull-Side-v1 | Pull a handle up sideways. | The handle is positioned for sideways pulling. |
| Stick-Push-v1 | Grasp a stick and push a box using the stick. | The stick and box are positioned randomly. |
| Stick-Pull-v1 | Grasp a stick and pull a box with the stick. | The stick and box are positioned randomly. |
| Basketball-v1 | Dunk the basketball into the basket. | The basketball and basket are positioned randomly. |
| Soccer-v1 | Kick a soccer ball into the goal. | The soccer ball and goal are positioned randomly. |
| Faucet-Open-v1 | Rotate the faucet counter-clockwise. | The faucet is positioned randomly. |
| Faucet-Close-v1 | Rotate the faucet clockwise. | The faucet is positioned randomly. |
| Coffee-Push-v1 | Push a mug under a coffee machine. | The mug and coffee machine are positioned randomly. |
| Coffee-Pull-v1 | Pull a mug from a coffee machine. | The mug and coffee machine are positioned randomly. |
| Coffee-Button-v1 | Push a button on the coffee machine. | The coffee machine's button is positioned randomly. |
| Sweep-v1 | Sweep a puck off the table. | The puck is positioned randomly on the table. |
| Sweep-Into-v1 | Sweep a puck into a hole. | The puck is positioned randomly on the table near a hole. |
| Pick-Out-Of-Hole-v1 | Pick up a puck from a hole. | The puck is positioned within a hole. |
| Assembly-v1 | Pick up a nut and place it onto a peg. | The nut and peg are positioned randomly. |
| Shelf-Place-v1 | Pick and place a puck onto a shelf. | The puck and shelf are positioned randomly. |
| Push-Back-v1 | Pull a puck to a goal. | The puck and goal are positioned randomly. |
| Lever-Pull-v1 | Pull a lever down 90 degrees. | The lever is positioned randomly. |
| Dial-Turn-v1 | Rotate a dial 180 degrees. | The dial is positioned randomly. |
| Bin-Picking-v1 | Grasp the puck from one bin and place it into another bin. | The puck and bins are positioned randomly. |
| Box-Close-v1 | Grasp the cover and close the box with it. | The box cover is positioned randomly. |
| Hand-Insert-v1 | Insert the gripper into a hole. | The hole is positioned randomly. |
| Door-Lock-v1 | Lock the door by rotating the lock clockwise. | The lock is positioned randomly. |
| Door-Unlock-v1 | Unlock the door by rotating the lock counter-clockwise. | The lock is positioned randomly. |

Our test tasks are the following: *assembly*, *basketball*, *bin picking*, *box close*, *button press topdown*, *button press topdown-wall*, *button press*, *button press wall*, *coffee button*, *coffee pull*, *coffee push*, *dial turn*, *disassemble*, *door close*, *door lock*, *door open*, *door unlock*, *drawer close*, *drawer open* , and *faucet close*.

## I    META-WORLD CLUSTERING ANALYSIS AND DISCUSSION

Simply put, our method works by having committee members which are innately specialized to specific tasks, as illustrated below. Here committee member 2 is specialized to *door open* and committee member 3 is specialized to *door close*. At the same time, committee member 2 performs

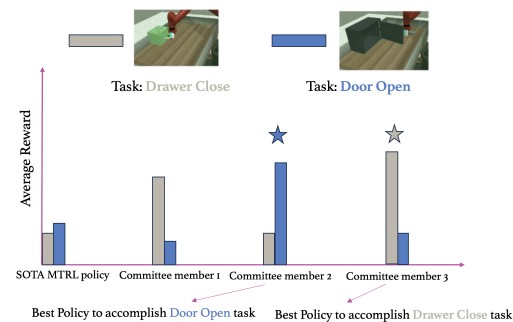

Figure 6: Performance on a single task across committee members compared to a MTRL policy.

*door close* poorly and committee member 2 performs *door open* poorly. A MTRL policy in trying to perform all tasks doesn't perform any particular task well. Our method will select committee member 2 for *door open* and committee member 3 for *door close*.

To understand if the parametrization discussed in section 3.4 produces suitable clusters we have applied PCA to PCA to a clustering of 10 tasks. We note that the window tasks and drawer tasks are close in task space. Additionally, the dynamics and goals of the *push* and *pick-place* tasks are nearly identical. *Window close* is close to *door open* as both these tasks have the agent needing to move to the horizontally to begin the task.

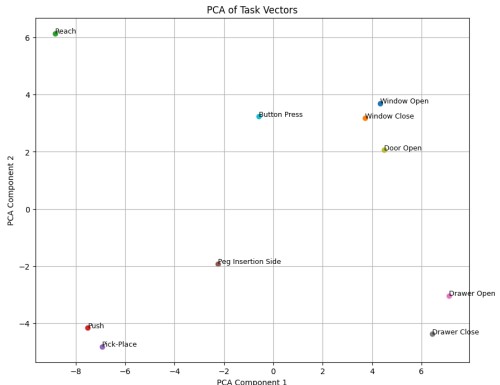

Figure 7: PCA for our parametrization described in 3.4.

