# OpenReview forum: "Learning Policy Committees for Effective Personalization in MDPs with Diverse Tasks"
_ICLR.cc/2025/Conference — Submitted to ICLR 2025_

### Official Review · Reviewer_UGnN · 2024-10-27

**Soundness:** 3
**Presentation:** 2
**Contribution:** 2
**Rating:** 5
**Confidence:** 4

**Summary:**

The paper built a meta-reinforcement learning framework that learns a set of policies such that at least one is near-optimal for most tasks that may be encountered. The solution is training a set of meta-models to achieve this goal.

**Strengths:**

The paper targets to address a big issue of meta-RL and multi-task RL, the trade-off between optimality and generalization when the tasks in the task distribution are diverse.

The literature review is complete and thoughtful.

The proposed method is justified and motivated by the theoretical results.

**Weaknesses:**

It is still unclear to me how the proposed method addresses the issue. In the context-based meta-RL method, the context of diverse tasks is learned, which can target different patterns for diverse tasks. What is the advantage of the proposed methods?

In the experiment, It is unfair to use the parametric task representation for the proposed method as the baselines of meta-RL do not use the information.

**Questions:**

Can the proposed method deal with the scenario without any parametric information about the tasks? If the tasks' parameter can be obtained, can we directly train a general policy conditional on the parameter that can deal with all tasks in the distribution?

How the proposed method can achieve zero-shot generalization for the new tasks? When a new task is given, how to figure out the task-specific policy from the candidates?

In the experiment, do the few-shot adaptations in all baselines require the same sample number?

---

> ### Author Response · Authors · 2024-11-21
>
> Thank you for your detailed reviews and thoughtful questions!
> > **Comment**: It is still unclear to me how the proposed method addresses the issue. In the context-based meta-RL method, the context of diverse tasks is learned, which can target different patterns for diverse tasks. What is the advantage of the proposed methods?
>
> **Response**: This is a really insightful point.  First, we note that in our experiments, VariBAD, RL2, CMTA, MOORE, and CARE are context-based methods which serve as our baselines. Thus, we provide strong empirical evidence that cleverly clustering the tasks into groups of semantically similar tasks yields significant boost in performance. Conceptually, while in principle one can learn policies that depend on arbitrary context (captured by task parameters), the practical issue is that the sample complexity of such approaches would require an enormous number of diverse training tasks, which we simply do not have in practice (or in Meta-World). In a world with a fairly limited set of tasks, grouping similar tasks together and learning a small set of distinct policies for each group performs far better, and our theoretical analysis provides a principled grounding for such an approach. We added some of this discussion in the revised Model section.
>
> > **Comment**: In the experiment, It is unfair to use the parametric task representation for the proposed method as the baselines of meta-RL do not use the information.
>
> **Response**: We wish to clarify that our multi-task baselines CMTA, MOORE, and CARE as well as the meta-RL baselines RL2 and VariBAD also depend on task-specific context representations. We view the fact that our approach can leverage *any* task representation as one of its key features in comparison to much prior work in which task representation is an inextricable part of the algorithm.
>
> > **Comment**:Can the proposed method deal with the scenario without any parametric information about the tasks?
>
> **Response**: Indeed, we devote Section 3.4 to this issue, and show that in many real tasks, such as Meta-World, we can use semantic (natural language) descriptions of the tasks and LLMs to obtain a useful task representation. Consequently, our method can be applied even to many practical non-parametric multi-task RL settings.
>
> > **Comment**: If the tasks' parameter can be obtained, can we directly train a general policy conditional on the parameter that can deal with all tasks in the distribution?
>
> **Response**: As we mention above, doing so successfully would require far more training tasks than is available in practical benchmark, such as Meta-World. Our key insight is that we can side-step this issue by leveraging natural similarity of many real-world tasks, while still accounting for diversity among tasks classes.
>
> > **Comment**: How the proposed method can achieve zero-shot generalization for the new tasks?
>
> **Response**: Our "zero-shot" experiments map to two potential use cases.  The first is analogous to LLM use cases where multiple responses to a single prompt can (and sometimes are) provided, with the constraint that the number of such simultaneous responses must be very small (e.g., $K\le 3$, as in all our experiments). The user then chooses the most helpful response. The second is when a set of new tasks arrives after training, and either human expertise (along with task descriptions, such as using natural language), or low-cost evaluation, can quickly determine which is the most appropriate policy.
>
> We deal with the latter setting more systematically in our "few-shot learning" experiments, which indeed demonstrate that one can quickly learn which policy among $K$ to use when facing new tasks.
>
> > **Comment**: Do the few-shot adaptations in all baselines require the same sample number?
>
> **Response**: Yes, to ensure a fair comparison, we keep the total number of samples during few-shot adaptation the same for all approaches.

---

> ### Author Response · Authors · 2024-11-24
> **Discussion period ending soon (Reviewer UGnN)**
>
> Dear Reviewer UGnN, the discussion period will end soon. We hope that you have had a chance to consider our responses.  Please let us know whether they have addressed your concerns, and if you have any further questions.
>
> Best,
>
> Authors

---

> > ### Comment · Reviewer_UGnN · 2024-11-24
> >
> > Thanks for your response.
> >
> > I still do not understand that "meta-RL baselines RL2 and VariBAD also depend on task-specific context representations". In my understanding, the context representation is automatically learned during the meta-training in the meta-RL baselines such as RL2. There is no need to request external information for the new task, e.g., the representation for the new task.
> >
> > Moreover, I am not clear why the proposed method can address the limitation of a small number of training tasks. When the number of training tasks is small, both the proposed method and meta-RL baselines suffer from the out-of-distribution issue when dealing with a new task.
> >
> > A quick question: how to do the experiments on the zero-shot setting for the meta-RL baselines?

---

> > > ### Author Response · Authors · 2024-11-24
> > > **Thank you for continued engagement**
> > >
> > > Dear Reviewer UGnN, we appreciate your quick response and continued engagement.
> > >
> > > > **Comment:** I still do not understand that "meta-RL baselines RL2 and VariBAD also depend on task-specific context representations". In my understanding, the context representation is automatically learned during the meta-training in the meta-RL baselines such as RL2. There is no need to request external information for the new task, e.g., the representation for the new task.
> > >
> > > **Response**: to clarify, we group multi-task and meta-RL methods together for this purpose, and the multi-task baselines are perhaps somewhat closer to what we do in this regard. In any case, the common way to capture context and task specifics is indeed learned during meta-training. We would generally expect that this can only improve performance, as it learns contextual representations that are optimized for generalization, whereas those specified exogenously are not.  We show that nevertheless, our approach that makes use of simple textual representations significantly outperforms such baselines.  Since both meta-RL and multi-task RL methods generally develop method-specific machinery for learning and using task representations, there is no obvious general-purpose way to simply "plug in" a new representation into an existing approach that would not essentially break the approach.  Thus, we provide what we felt was the most fair comparison in this regard.
> > >
> > > > **Comment**: Moreover, I am not clear why the proposed method can address the limitation of a small number of training tasks. When the number of training tasks is small, both the proposed method and meta-RL baselines suffer from the out-of-distribution issue when dealing with a new task.
> > >
> > > **Response**: This is actually what our theoretical analysis shows.  For example, Theorem 4 connects the finite-sample training to generalization in the sense of the $(\epsilon,1-\delta)$-parameter-cover concept that we have defined, and Theorem 6 connects the concept of $(\epsilon,1-\delta)$-cover to few-shot learning guarantees. *Notably, neither result depends on the dimension of the parametric representation of tasks*.
> > >
> > > > **Comment**: how to do the experiments on the zero-shot setting for the meta-RL baselines?
> > >
> > > **Response**: we simply omit the typical "few-shot" training stage for these baselines (as for all the other baselines, and our approach), and apply the trained "initial" policies directly.

---

> > > > ### Comment · Reviewer_UGnN · 2024-11-25
> > > >
> > > > Thanks for the reply.
> > > >
> > > > I still think the setting of the paper is not meta-RL and it only considers multi-task RL. The zero-shot experiments compared with the meta-RL is unfair, because it does not assume that a human expert can select the policy for the meta-RL. So I retain my score.

---

> > > > > ### Author Response · Authors · 2024-11-25
> > > > >
> > > > > Thank you for the comment.  We wish to clarify that our key comparison with meta-RL is in the few-shot setting, where our performance is significantly better than all the meta-RL baselines.  Do you have specific concerns about our few-shot evaluation, which we felt would address your comment?

---

> > > > > > ### Author Response · Authors · 2024-12-01
> > > > > > **Discussion period ending**
> > > > > >
> > > > > > Dear reviewer UGnN,
> > > > > >
> > > > > > The extended discussion period is nearing its end, and we hope that the reviewer can take our latest comment into consideration.
> > > > > >
> > > > > > > **Comment**: I still think the setting of the paper is not meta-RL and it only considers multi-task RL.  The zero-shot experiments compared with the meta-RL is unfair, because it does not assume that a human expert can select the policy for the meta-RL.
> > > > > >
> > > > > > **Response**: The training and zero-shot results (Figure 1 and Table 2) are indeed focusing on the multi-task RL setting.  However, our *few-shot learning* results in Tables 1 and 3 are for the meta-RL setting.  For these results, we automatically choose a policy to implement from the committee using few-shot policy evaluation, and the setup (including the total number of training iterations being split among the policies in the committee) provides for a fair "apples-to-apples" comparison between our approach and meta-RL baselines.

---

### Official Review · Reviewer_GRXu · 2024-10-31

**Soundness:** 3
**Presentation:** 2
**Contribution:** 2
**Rating:** 6
**Confidence:** 3

**Summary:**

This paper studies multi-task reinforcement learning and few-shot learning.

For multi-task learning, authors consider learning a set of policies called a policy committee to handle multiple tasks. For the case of identical transition dynamics and different rewards across tasks, the authors further provide theoretical results to show they can find good coverage for the parametric model and thus good coverage for all optimal policies. For more general different transition dynamics cases, the authors show some empirical results.

For few-shot learning, the authors provide some sample complexity bound with certain assumptions.

**Strengths:**

(1) The authors carefully consider different scenarios for multi-task learning (low-dimensional or high-dimensional, parametric or non-parametric), and provide the algorithms accordingly (some with theoretical guarantee, some without).

(2) The authors provide empirical results for cases hard to analyze.

**Weaknesses:**

(1) In the multi-task case with identical transition kernels and different rewards. Another straightforward way is to commit reward-free RL algorithm [1] ([1] only considers tabular RL, but there are follow-ups considering function approximation via linear and low-rank MDPs), where the agent can directly learn the transition kernel well enough to deal with all kinds of reward. A line of work also use this methods to study multi-task [2,3]. I wonder how your theoretical results compare with this work under the scenario with identical transition kernels and different rewards.

(2) In Gradient-Based Coverage, it seems that your cluster representation can only be selected from tasks set $T$, which is different from the Greedy intersection algorithm, and your theorem 5 also only holds for this case. Intuitively, the error gap between cluster representation and task w.r.t. this representation will grow larger for Gradient-Based Coverage than the Greedy intersection algorithm. I wonder why you chose different regimes here, and it seems not reasonable for Gradient-Based Coverage.






[1] Jin C, Krishnamurthy A, Simchowitz M, Yu T. Reward-free exploration for reinforcement learning. InInternational Conference on Machine Learning 2020 Nov 21 (pp. 4870-4879). PMLR.

[2] Cheng Y, Feng S, Yang J, Zhang H, Liang Y. Provable benefit of multitask representation learning in reinforcement learning. Advances in Neural Information Processing Systems. 2022 Dec 6;35:31741-54.

[3] Agarwal A, Song Y, Sun W, Wang K, Wang M, Zhang X. Provable benefits of representational transfer in reinforcement learning. InThe Thirty Sixth Annual Conference on Learning Theory 2023 Jul 12 (pp. 2114-2187). PMLR.

**Questions:**

Please see above.

---

> ### Author Response · Authors · 2024-11-21
>
> Thank you for the detailed reviews and thoughtful feedback!
> > **Comment**: I wonder how your theoretical results compare with transition-kernel learning RL work [1-3] under the scenario with identical transition kernels and different rewards.
>
> **Response**: Thank you for your insightful question! The main differences are that this prior work (and related papers) typically makes significant stronger assumptions. A common example is that the action space is typically assumed to be finite. The papers that do not assume this require instead linearity (of both the dynamics and rewards), and/or make assumptions (such as low-rank) on the transition model. We do not assume linearity, make no assumptions about the nature of the dynamics, and allow actions to be either discrete or in a continuous vector space (none of our results depend on the size or dimension of state or action space). Thus, one of our key results in few-shot generalization depends only on the number of policies $K$ (i.e., sample complexity is linear in $K$, along with other problem-specific features), but has no dependence on state or action space dimension.
>
> > **Comment**: In Gradient-Based Coverage, it seems that your cluster representation can only be selected from tasks set $T$.
>
> **Response**: We wish to clarify that our gradient-based coverage approach optimizes the parameter vector *in the space of all feasible task parameter vectors*, not merely restricted to $T$. The key advantage of this algorithm over Greedy Intersection is that it scales far better in the dimension of the parameter space.

---

> ### Author Response · Authors · 2024-11-24
> **Discussion period ending soon (Reviewer GRXu)**
>
> Dear Reviewer GRXu, the discussion period will end soon. We hope that you have had a chance to consider our responses.  Please let us know whether they have addressed your concerns, and if you have any further questions.
>
> Best,
>
> Authors

---

> > ### Comment · Reviewer_GRXu · 2024-11-26
> >
> > Thanks for your response.
> >
> > My concerns are addressed and I have raised my scores.

---

### Official Review · Reviewer_GmNe · 2024-11-03

**Soundness:** 2
**Presentation:** 3
**Contribution:** 2
**Rating:** 6
**Confidence:** 4

**Summary:**

This paper presents a novel method for multi-task reinforcement learning with diverse tasks. Instead of using a single policy to act on a random draw of a task (as common in meta-rl and multi-task rl), the authors suggest clustering the task based on the task parameters and learning a different policy for each cluster. First, theoretical motivations were given for the connection between the clustering problem to the original problem, the clustering problem itself, and a proposed few-shot adaptation using the policy committee. Secondly, the authors performed an empirical study on two common benchmark suits while comparing to multiple meta and multi-task rl methods.

**Strengths:**

1. Effectively learning a committee of policies instead of one policy for all tasks has a big potential to help in real-world application of multi-task rl, though a more thorough analysis of this is lacking (see weakness 1)

2. Overall the presentation of the paper is good, the notations are clear and the ideas presented are well-motivated.

3. The Related Work section is well-articulated and contains a broad spectrum of related topics.

4. The Metaworld benchmark is very popular in the meta rl and multi-task rl literature and the authors compared to multiple common baselines in the fields.

**Weaknesses:**

1. While the theorems shown in the paper motivate **how** to find a cover for a set/distribution of tasks, there is little to no discussion on **why** should it help compared to a single policy. The only mention I found regarding this question is the paragraph in lines 153-159. Theoretical analysis and motivation on when a policy committee should help compared to a single policy is missing. For example, in the mentioned paragraph the authors claim a policy committee should help when faced with outlier tasks, does theory support this? Is this the only case a policy committee should be beneficial? If so, why should it have superior results in the Mujoco benchmark where there aren’t any outliers?

2. A big limitation that was not discussed throughout the paper, both in the theoretical and empirical parts is the assumption of access to the parametric space of tasks and the mapping from that space to the space of MDPs. The access to the parametric space is needed to perform the clustering itself and the access to the mapping is needed in the theory part in order to map between the covering parameters to MDPs to learn the policies, which in turn will consist of the committee. Both of these assumptions are substantial and a discussion of them and ways to alleviate them can make the paper stronger. One possible way to do so is to use a learned parametric representation as done in previous works (e.g. [2, 3]).
One of the biggest (not discussed) advantages of the suggested “naive” Greedy Elimination Algorithm is that it doesn’t require access to the mapping function as it only uses existing tasks.

3. Some of the settings in the theoretical part did not match the ones in the empirical study. The main difference (which was not discussed) is that in the empirical study the authors suggested that each policy in the committee will be trained on all tasks in the respective cluster, whereas in the theoretical analysis, the authors assumed each policy is the optimal one for a specific task. This modification is important as in some tasks (e.g. pick and place), the Lipshitz assumption in Lemma 1 will not hold and without the modification, I would expect the method to fail. Theoretical guarantees for the settings in the empirical part would strengthen the paper.

4. I found certain sections of the paper somewhat unclear, particularly the Greedy Intersection Algorithm and the latter part of Section 3.3. Further, while the theory is well formulated overall, the readability could be improved if the authors provided motivation before introducing key definitions and lemmas. Additionally, unifying the notation for meta-RL and multi-task RL (i.e., $\Gamma$ and $T$) as the latter representing a discrete case of the former, can help readability.

5. The empirical study can be extended to support some of the claims in the paper better, for example:

    a. One of the main practical algorithmic novelties in the paper is the task clustering algorithm, the only ablation on this is the result in Figure 2(a), which shows a marginal improvement over KMeans. A further investigation with more clustering algorithms and ablation over the design choices of the clustering algorithm can improve the soundness of the proposed method.

    b. The authors used popular meta-rl baselines, but more recent baselines are missing, e.g [1, 2, 3]

    c. To make a fair comparison to VariBAD/MOORE the authors should’ve compared the total number of parameters in the policy committee and the baseline. Adding a baseline with the same number of parameters or a policy committee where the tasks are randomly split between the committee can make the claims stronger

[1] AMAGO: Scalable In-Context Reinforcement Learning for Adaptive Agents

[2] Rimon et al. - Meta Reinforcement Learning with Finite Training Tasks -- a Density Estimation Approach

[3] Lee et al. Improving generalization in meta-rl with imaginary tasks from latent dynamics mixture

**Questions:**

1. Typos:

    a. Line 155 - to to -> to

    b. Line 143 - denote an optimal policy -> denote the value of an optimal policy

    c. Line 215 - task $\pi_i$ -> task $\tau_i$

    d. Line 516 - Figure 5.3(a) -> Figure 2(a)

2. The first two plots Figure 1 seem wrong, they don’t fit the claims made in the paper, as it seems that the new algorithm doesn’t perform better than the baselines.

3. Did you choose $\epsilon$ and $K$ by trying different values and picking the best one? Should the distribution of tasks affect these values, and if so how?

4. Is there a reason not to increase $K$? I would expect better and better results as $K$ increases.

5. In the encoding of the Meta World tasks using the language model, did you also use the goal location in the task description?

6. In the Meta World experiment, what were the results of your clustering algorithm? Was there any semantic meaning for the clustering?

7. Did you tune the hyperparameters for the in-cluster policies or did you use the same hyperparameters as VariBAD/MOORE?

8. In VariBAD they reported higher results on HalfCheetah, is it a different environment?

9. How many seeds did you use in your empirical studies?

10. In Section 5.3 you show the results of PACMAN with K=1 but the results are very different than VariBAD/MOORE. Isn’t PACMAN with K=1 identical to the original algorithms?

11. Why did you choose to train in the Mujoco baseline for just 1.2e7 steps? In VariBAD for example it seems they trained for much longer.

---

> ### Author Response · Authors · 2024-11-21
>
> Thank you for your detailed reviews and thoughtful questions! Please check on our new revision.
>
>
> > **Comment**: No discussion on why covering should help compared to a single policy...  For example, in the mentioned paragraph the authors claim a policy committee should help when faced with outlier tasks, does theory support this?
>
> **Response**: We expanded this discussion in the revised version. In particular, using the "outlier" terminology was misleading. What we really mean is that when the set of tasks is highly diverse, it is unlikely that a single policy will effectively solve all of them very effectively, as fundamnetally different skills may be required for different sets of tasks. Consequently, what can often happen in practice is that different tasks confound one another's training. That the key insight that gives rise to our approach. Moreover, if tasks are represented parametrically, this representation can be used to infer the extent to which this diversity is present (in terms of how similar, or how spread out, the tasks are in the parameter space). Our theoretical results capture this by translating it into a formal notion of "task cover", and using this concept to obtain clustering algorithms that group similar tasks together (i.e., which presumably require similar sets of skills to accomplish) and learn a distinct model that is most appropriate for a given group of tasks. While in principle it is possible to learn a single policy that takes parametric representations of tasks as an input to account for such diversity, in practice this requires a prohibitive number of sample tasks, especially if task representations are high-dimensional and tasks very diverse. Our approach thus allows a very efficient use of a relatively small sample of tasks, as demonstrated in our experiments (for example the MOORE and CARE baseline use context information).
>
> > **Comment**: A big limitation that was not discussed throughout the paper, both in the theoretical and empirical parts is the assumption of access to the parametric space of tasks and the mapping from that space to the space of MDPs. The access to the parametric space is needed to perform the clustering itself and the access to the mapping is needed in the theory part in order to map between the covering parameters to MDPs to learn the policies, which in turn will consist of the committee. Both of these assumptions are substantial and a discussion of them and ways to alleviate them can make the paper stronger. One possible way to do so is to use a learned parametric representation as done in previous works (e.g. [2, 3]).
>
> **Response**: This is a great point, and one that we deal with practically in Section 3.4, where we suggest that for many real non-parametric multi-task environments, we can extract a representation (embedding) using natural language descriptions and pre-trained LLMs. This is conceptually inline with [2,3], but simpler, and (as we show) quite effective in practice in the case of Meta-World.
>
> >**Comment**: One of the biggest (not discussed) advantages of the suggested “naive” Greedy Elimination Algorithm is that it doesn’t require access to the mapping function as it only uses existing tasks.
>
> Such feature extraction function is actually important even for Greedy Elimination, since the coverage criterion that this (and all other) algorithms we present relies on is representation similarity. Our key practical idea is that once we have identified the clusters, we can execute RL on the set of tasks comprising each cluster. We now made this point more explicitly in Section 3.3.
>
> > **Comment**: Some of the settings in the theoretical part did not match the ones in the empirical study. The main difference (which was not discussed) is that in the empirical study the authors suggested that each policy in the committee will be trained on all tasks in the respective cluster, whereas in the theoretical analysis, the authors assumed each policy is the optimal one for a specific task. This modification is important as in some tasks (e.g. pick and place), the Lipshitz assumption in Lemma 1 will not hold and without the modification, I would expect the method to fail. Theoretical guarantees for the settings in the empirical part would strengthen the paper.
>
> **Response**: This is a very insightful point. Indeed, our theoretical analysis is slightly stylized in this sense. What we show is that it nevertheless provides us with useful practical algorithms that outperform the state of the art in the complex benchmark setting of Meta-World, both in terms of multi-task RL (in-sample performance) and meta-RL (generalization and few-shot learning). In other words, our theoretical analysis provides a principled grounding for the final practical algorithms that we develop and use.

---

> > ### Author Response · Authors · 2024-11-21
> >
> > > **Comment**: I found certain sections of the paper somewhat unclear, particularly the Greedy Intersection Algorithm and the latter part of Section 3.3. Further, while the theory is well formulated overall, the readability could be improved if the authors provided motivation before introducing key definitions and lemmas.
> >
> > **Response**: We edited our discussion of the Greedy Intersection to improve readability. We also removed the latter part of Section 3.3.
> >
> >
> >
> > > **Comment**: Unifying the notation for meta-RL and multi-task RL as the latter representing a discrete case of the former, can help readability.
> >
> > **Response**: This is indeed what we tried to do in the model (as the explicit distinction between MT-MDP and FS-MT-MDP). Are there specific things that we can further improve in this regard?
> >
> >
> > >**Comment**: A further investigation with more clustering algorithms and ablation over the design choices of the clustering algorithm can improve the soundness of the proposed method.
> >
> > **Response**:
> >
> > Thank you for the suggestion! We added a comparison to three additional clustering algorithms:
> > 1. Gaussiam Mixture Models
> > 2. DBSCAN, with hyperparameters tuned to obtain the target number of clusters.
> > 3. Random clustering, to ablate that our improvment is not due to the statistical consequence of simply adding more policies
> >
> > The new evaluation curve is added to the appendix. At the high level, we find that KMeans++ (which was our original baseline that we refer to as kmeans) is by far the most competitive baseline.
> >
> > We also note that the improvement compared to KMeans++  is non-trivial: as the table below shows, after 1.2e7 frames, we exhibit a 20% improvement in performance (the difference looks smaller on the plot since both achieve policies that are dramatically better than random)
> >
> >
> > | Test Zero-shot | Ours   | KMeans   | GMM    | DBScan|Random   |
> > |-------|--------|--------|--------|--------|--------|
> > |12000000 Frames| -74.20 | -89.50 |-202.55  | -213.43  | -258.07 |
> >
> > In addition, if one looks at the distribution of performance (Fig. 4, left in Appendix G.2.1), the performance distribution shifts significantly to the right for our method compared to KMeans++; the difference between averages seems closer due to a small number of outliers.
> >
> >
> > >**Comment**: The authors used popular meta-rl baselines, but more recent baselines are missing, e.g [1, 2, 3]
> >
> > **Response**:
> >
> > Thank you for the suggestion! Of the three, only AMAGO [1]---which is the most recent approach---has been evaluated in Meta-World (the other two do not appear to readily work in that environment). We implemented this baseline and added it to our results (updated Fig. 1 and Table 3 in the revised draft).
> >
> > As shown in the table below, our approach significantly outperforms AMAGA as well (it is not the most competitive baseline, although it is among the best meta-RL baselines).
> >
> > |    Train Zero Shot   | AMAGO    | PACMAN |
> > |-------|--------|--------|
> > | 500K Steps| .04  | .54   |
> > |1M Steps| .05  | .55   |
> >
> > |    Test Zero Shot    | AMAGO    | PACMAN|
> > |-------|--------|--------|
> > | 500K Steps| .10  | .45   |
> > |1M Steps| .14  | .45   |
> >
> > |    Few-Shot Eval    | AMAGO    | PACMAN|
> > |-------|--------|--------|
> > | 6k updates| .08  | .53   |
> > |12k updates| .09  | .60   |
> >
> >
> >
> > >**Comment**: To make a fair comparison to VariBAD/MOORE the authors should’ve compared the total number of parameters in the policy committee and the baseline.
> >
> > **Response**: This is an insightful point. Our current focus is on learning efficacy, rather than representational complexity; indeed, different baseline approaches vary significantly in the number of parameters, as this is not typically a primary consideration in prior literature, particularly in the context of the challenging Meta-World benchmark.
> >
> > > **Comment:** Typo and Figure 1 issue
> >
> > **Response**: Thank you for pointing this out! The legend for two of the left plots was missing. We fixed this issue in the revision. Indeed, our approach outperforms all baselines.
> >
> >
> >
> > > **Comment:** Did you choose $\epsilon$ and $K$ by trying different values and picking the best one? Should the distribution of tasks affect these values, and if so how?
> >
> > **Response**: We tried a small number of combinations for $\epsilon$ and $K$, using the distribution of distances between tasks to identify a reasonable value for $\epsilon$. We generally take $K$ to be constrained by other factors, for example, practical considerations in the number of RL policies we can use simultaneously, as well as few-shot learning, where sample complexity increases with $K$ (Theorem 6 in Section 4). A simple way to identify $K$ is also to check the fraction of in-sample tasks covered as a function of $K$, so that 100% coverage provides a natural upper bound.

---

> > > ### Author Response · Authors · 2024-11-21
> > >
> > > >**Comment:** Is there a reason not to increase $K$ ? I would expect better and better results as $K$ increases.
> > >
> > > **Response**: We further conducted our ablation experiment with K=4 as suggested; the results are provided in the table below for reference, and also added to the main draft in Figure 2. What we find (and is somewhat expected) is that increasing $K$ beyond 3 is actually harmful for performance.  This is because when we split the tasks into more clusters, each cluster has fewer source tasks to train on, resulting in lower generalization performance.
> > >
> > > |    Train, $\epsilon=.7$   | K=1    | K=2    | K=3    | K=4    |
> > > |-------|--------|--------|--------|--------|
> > > | 500K Steps| .18  | .45  | .51  | .32  |
> > > |1M Steps| .23  | .45  | .53  | .35  |
> > >
> > > |    Test, $\epsilon=.7$   | K=1    | K=2    | K=3    | K=4    |
> > > |-------|--------|--------|--------|--------|
> > > | 500K Steps| .16  | .40  | .48  | .30  |
> > > |1M Steps| .23  | .45  | .55  | .35  |
> > >
> > > > **Comment:** For meta-world, did you also use the goal location in the task description?
> > >
> > > **Response**: The objective of the task is part of the textual description, but *not necessarily the precise goal location*. The detailed task descriptions are provided in Appendix H.
> > >
> > > > **Comment:** In the Meta World experiment, what were the results of your clustering algorithm? Was there any semantic meaning for the clustering?
> > >
> > > **Response**: Yes, added a plot of the PCA for the clusters in Appendix I. We noted that drawer open and draw close were close, as well as window open and window close. We also found push and pick place close, which is what we would expect, as these tasks both involve picking and moving an object to a goal, with the difference the push does not pick it up and pick place does (see https://meta-world.github.io/figures/mt10.gif for reference).
> > >
> > > > **Comment:** Did you tune the hyperparameters for the in-cluster policies or did you use the same hyperparameters as VariBAD/MOORE?
> > >
> > > **Response**: We used the same set of hyperparameters for all clusters.
> > >
> > > > **Comment:** In VariBAD they reported higher results on HalfCheetah, is it a different environment?
> > >
> > > **Response**: The reason that the original VariBAD results are better is that they simulate a much larger number of training tasks (millions), whereas we only use a training set of 100 tasks for all approaches in MuJoCo, as our key goal is to demonstrate that our approach is far more sample efficient.
> > >
> > >
> > >
> > > > **Comment:** How many seeds did you use?
> > >
> > > **Response**: We used 3 seeds in Meta-World.  In MuJoCo experiments, we used 10 seeds for the baselines as they have much higher variance than our approach.
> > >
> > > >  **Comment:** Isn’t PACMAN with K=1 identical to the original algorithms?
> > >
> > > **Response**: This is a really insightful question. The reason that PACMAN with $K=1$ is not identical to the original algorithm is that the clustering approach will only use a single cluster that yields an $\epsilon$-cover for the largest set of tasks. It then trains only on this subset of all tasks, whereas the original algorithm would just train on all of the tasks.
> > >
> > > > **Comment:** Why did you choose to train in the Mujoco baseline for just 1.2e7 steps?
> > >
> > > **Response**: We actually ran the same number of steps as in the original paper, but performance of the baselines does not meaningfully increase beyond 1.2e7 steps.

---

> > > > ### Comment · Reviewer_GmNe · 2024-11-23
> > > > **Follow-up Questions**
> > > >
> > > > I appreciate the effort and modifications the authors made during the rebuttal as well as answering all of my questions. I think the added discussions and experiments make the paper much stronger. I’m inclined to reconsider my score, but would appreciate the authors’ comments on the follow-up questions first.
> > > >
> > > > Regarding Weakness #2: I thank the authors for the clarification. Regarding Section 3.4 - if I understand correctly, you still assume access to the textual representation of each task, right? Further, I think a discussion on those assumptions (at least in the theoretical parts) is important and missing. The added discussion in Section 3.3 is not enough in my opinion.
> > > > For example -
> > > > 1. “They may help with the generalization as the previous multi-task learning literature has pointed out” - make this explicit (i.e which prior works and what are the claims you’re talking about)
> > > > 2. “Moreover, training policies on imaginary tasks may not be feasible (in which case we can use the Greedy Elimination approach)” - what do you mean by imaginary tasks? Why it may not be feasible? Why Greedy Elimination solves this problem.
> > > >
> > > > Regarding Weakness #4:  My thoughts were that you could drop the distinction between the discrete and continuous cases and just discuss the continuous case within the Lemmas, which should help readability. This is not a must though in my opinion.

---

> > > > > ### Author Response · Authors · 2024-11-24
> > > > > **Follow-up Response**
> > > > >
> > > > > We are truly grateful to the reviewer for spending the time and effort to review our revised draft, and engage in further discussion. We uploaded an updated draft in which we aim to address the weaknesses highlighted, as we elaborate on below.
> > > > >
> > > > > >**Comment:** Regarding Section 3.4 - if I understand correctly, you still assume access to the textual representation of each task, right? Further, I think a discussion on those assumptions (at least in the theoretical parts) is important and missing.
> > > > >
> > > > > **Response**: Fundamentally, what we assume is that there is a feature extractor that can map an arbitrary task $\tau$ to a feature representation $\theta$. In our MuJoCo experiments, such parametric representations are direct, while in Meta-World, we do indeed rely on having a textual representation of each task. Because in principle we can use *any* feature representation of tasks, we prefer not to narrow down the scope in making textual description an explicit assumption. Instead, we revised the begining of Section 3.4 as follows:
> > > > >
> > > > > "Our approach assumes that tasks are parametric, so that we can reason (particularly in the clustering step) about parameter similarity. Many practical multi-task settings, however, are non-parametric, so that our algorithmic framework cannot be applied directly. In such cases, our approach can make use of any available method for extracting a parametric representation of an arbitrary task $\tau$. For example, it is often the case that tasks can be either described in natural language. We propose to leverage this property and use text embedding (e.g., from pretrained LLMs) as the parametric representation of otherwise non-parametric tasks, where this is feasible."
> > > > >
> > > > > We hope that this revision better clarifies the scope of applicability of our approach.
> > > > >
> > > > > >**Comment:** Section 3.3: “They may help with the generalization as the previous multi-task learning literature has pointed out” - make this explicit (i.e which prior works and what are the claims you’re talking about)
> > > > >
> > > > > **Response**:
> > > > > That is a good point, thank you. We added the references, and elaborated as follows:
> > > > >
> > > > > "As demonstrated empirically in the multi-task RL literature, using multiple tasks to learn a shared representation facilitates generalization (effectively enabling the model to learn features that are beneficial to all tasks in the cluster) (Sodhani et al., 2021; Sun et al., 2022; Yang et al., 2020b)."
> > > > > >**Comment:** Moreover, training policies on imaginary tasks may not be feasible (in which case we can use the Greedy Elimination approach)” - what do you mean by imaginary tasks? Why it may not be feasible? Why Greedy Elimination solves this problem.
> > > > >
> > > > > **Response**:
> > > > > We mean that for settings such as Meta-World, it is not possible to generate an arbitrary task for a given $\theta$, so that just using the representative $\theta$ for a cluster may not even be feasible. In any case, we ultimately removed this sentence which makes our discussion needlessly confusing for a reader.
> > > > > >**Comment:** My thoughts were that you could drop the distinction between the discrete and continuous cases and just discuss the continuous case within the Lemmas, which should help readability. This is not a must though in my opinion.
> > > > >
> > > > > **Response**:
> > > > >
> > > > > Yes, this is a great suggestion! We implemented it in the revised draft (and instead just added a footnote that our results also work for the FS-MT-MDP setting).

---

> > > > > > ### Comment · Reviewer_GmNe · 2024-11-24
> > > > > >
> > > > > > I thank the authors for the modifications and interesting discussion.
> > > > > > While I still have major concerns regarding the theoretical part, the improved empirical study led me to raise my score to 6 and recommend acceptance. I chose not to raise my score further as some of my main concerns remain unresolved.
> > > > > > 1. Limited to no theoretical analysis of **why** and **when** a committee of policies should beat a single policy approach
> > > > > > 2. While the authors did add discussions on Weakness #2, it still stands as one of the biggest limitations of the proposed approach. If the authors would have, for example, performed their approach on a learned parametric representation (similar to [1,2,3]), that could have made this work much stronger and more applicable in practice. Even a discussion on this as future work could have helped position the paper better.
> > > > > > 3. While I appreciate the authors' response regarding Weakness #3, this remains a major concern in my opinion. There is a big gap between the assumptions in the theoretical part and the empirical part (as discussed in my original comment), which I think can be (at least partially) solved by further theoretical analysis.
> > > > > >
> > > > > >
> > > > > > [1] Zintgraf et al - VariBAD: A very good method for Bayes-Adaptive Deep RL via Meta-Learning
> > > > > >
> > > > > > [2] Rimon et al. - Meta Reinforcement Learning with Finite Training Tasks -- a Density Estimation Approach
> > > > > >
> > > > > > [3] Lee et al. Improving generalization in meta-rl with imaginary tasks from latent dynamics mixture

---

> > > > > > > ### Author Response · Authors · 2024-11-24
> > > > > > >
> > > > > > > We thank the reviewer for the continued discussion, and for recommending acceptance.  We will add a discussion of these limitations as important directions for future work in the revised version (we suspect that the theoretical analysis alone that these entail will make for an interesting independent contribution, or perhaps even several).

---

### Official Review · Reviewer_WKRZ · 2024-11-04

**Soundness:** 3
**Presentation:** 2
**Contribution:** 3
**Rating:** 6
**Confidence:** 2

**Summary:**

The paper studied multi-task reinforcement learning and proposed an algorithm that is able to identify a set of policies that includes the optimal policies of majority tasks. The paper theoretically characterized the performance of the algorithm and also conducted experiments to demonstrate the effectiveness of the proposed methods.

**Strengths:**

1. The proposed method of learning a committee of policies for different tasks is novel, because it provides a potential solution to multi-task RL by enabling more adaptability.
2. The investigation is thorough by providing both theoretical and experimental results.

**Weaknesses:**

1. The presentation could be further improved, especially in section 3. For example, while the title of sec 3.2 is 'Clustering', there is no explicit description of how to do cluster in the following context. I would assume the construction of $C$ in definition 2 should be the clustering method. Please correct me if I am wrong. It is better to use pseudo-code to highlight the steps.
2. Another weakness has been discussed a little by authors. That is, the algorithm relies on task representations.

**Questions:**

The impact of hyper-parameters on the performance can be further explained. For example, does a larger committee policies ($K$) always yield better results? How should we choose $K$ in practical applications?

---

> ### Author Response · Authors · 2024-11-21
>
> Thank you for your detailed reviews and thoughtful questions! Please check on our new revision.
>
> > **Comment**: no explicit description of how to do cluster in the following context (Sec 3.2).
>
> **Response**: This section is devoted to presenting our algorithmic approaches for clustering. We provide the pseudocode (Algorithm 1, in the Appendix) for our first main contribution, which is the Greedy Intersection clustering algorithm. Greedy elimination simply iteratively chooses a single $\theta$ from the set of all tasks which covers the most tasks not previously covered. Finally, the gradient-based coverage method uses conventional gradient descent to minimize the objective in Equation (2).
>
> > **Comment**: The algorithm relies on task representations.
>
> **Response**: Indeed, this is an important assumption in our work, and we devote Section 3.4 to this issue. In particular, we show that the assumption is not as strong as it first appears to providing an approach (with further details in the Appendix) for using pretrained LLMs to obtain parametric representations (embeddings) from textual descriptions of non-parametric tasks in Meta-World, which is a major benchmark for multi-task RL and meta-RL.
>
> > **Comment:**: The impact of hyperparameters $K$ and $\epsilon$ on performance.
>
> We added further ablations to evaluate the impact of these for Meta-World and updated Fig. 2 as well as added new results in Appendix Section G.2.2.
>
> |    Train, $\epsilon=.7$   | K=1    | K=2    | K=3    | K=4    |
> |-------|--------|--------|--------|--------|
> | 500K Steps| .18  | .45  | .51  | .32  |
> |1M Steps| .23  | .45  | .53  | .35  |
>
> |    Test, $\epsilon=.7$   | K=1    | K=2    | K=3    | K=4    |
> |-------|--------|--------|--------|--------|
> | 500K Steps| .16  | .40  | .48  | .30  |
> |1M Steps| .23  | .45  | .55  | .35  |
>
> An interesting observation from our experiments is that increasing $K$ beyond what already provides full coverage for a given $\epsilon$ significantly harms performance. The reason is that it is useful to have sufficient tasks for each cluster to enable effective and robust per-cluster training.
>
>
> |    All tasks, $K=2$   | $\epsilon=.4$    | $\epsilon=.7$    | $\epsilon=1$    |
> |-------|--------|--------|--------|
> | 500K Steps| .05  | .28  | .29   |
> |1M Steps| .05  | .31  | .40  |
>
> We find that increasing $\epsilon$ to cover more tasks can also improve performance (for a similar reason that increasing $K$ may not, as higher $\epsilon$ can ensure that we do not end up with clusters with too few tasks). Of course, for sufficiently high $\epsilon$, only a single cluster will emerge, so this, too induces an interesting tradeoff.

---

> > ### Comment · Reviewer_WKRZ · 2024-11-24
> >
> > Thanks for the reply. I have no further question and decide to retain my score.

---

### Meta-Review · Area_Chair_Bt7h · 2024-12-22

**Metareview:**

This paper presents a new method for multi-task reinforcement learning with diverse tasks. Instead of using a single policy to act on a random draw of a task, the authors suggest clustering the task based on the task parameters and learning a different policy for each cluster. First, theoretical motivations were given for the connection between the clustering problem and the original problem, the clustering problem itself, and a proposed few-shot adaptation using the policy committee. Secondly, the authors performed an empirical study on two common benchmark suits while comparing them to multiple meta and multi-task RL methods.

All reviewers and the AC agree that this paper has good potential for future research. However, given the limitations of the paper, the AC recommends rejection.

**Additional Comments On Reviewer Discussion:**

The main concerns are: 1) access to a parametric space of tasks, 2) the gap between theory and empirical findings, 3) no formal theoretical separation that shows the committee of policies beast a single policy approach. These concerns are somewhat addressed in the rebuttal but not fully.

---

### Decision · Program_Chairs · 2025-01-22

Reject